# A robotic sensory system with high spatiotemporal resolution for texture recognition

Ningning Bai[1,2,5], Yiheng Xue [3,5], Shuiqing Chen[3], Lin Shi[1], Junli Shi[1], Yuan Zhang[1], Xingyu Hou[1], Yu Cheng[1], Kaixi Huang[1], Weidong Wang [2], Jin Zhang[3], Yuan Liu[4] & Chuan Fei Guo [1] ✉

Humans can gently slide a finger on the surface of an object and identify it by capturing both static pressure and high-frequency vibrations. Although modern robots integrated with flexible sensors can precisely detect pressure, shear force, and strain, they still perform insufficiently or require multi-sensors to respond to both static and high-frequency physical stimuli during the interaction. Here, we report a real-time artificial sensory system for high-accuracy texture recognition based on a single iontronic slip-sensor, and propose a criterion—spatiotemporal resolution, to corelate the sensing performance with recognition capability. The sensor can respond to both static and dynamic stimuli (0-400 Hz) with a high spatial resolution of 15 μm in spacing and 6 μm in height, together with a high-frequency resolution of 0.02 Hz at 400 Hz, enabling high-precision discrimination of fine surface features. The sensory system integrated on a prosthetic fingertip can identify 20 different commercial textiles with a 100.0% accuracy at a fixed sliding rate and a 98.9% accuracy at random sliding rates. The sensory system is expected to help achieve subtle tactile sensation for robotics and prosthetics, and further be applied to haptic-based virtual reality and beyond.

Robotic technologies have a growing demand for tactile sensation to enable friendly interaction between robots and their surroundings[1–6]. This functionality is often realized using artificial sensory systems based on flexible tactile sensors. Existing flexible tactile sensors mostly focus on the precise detection of physical stimuli, including pressure, shear force, and strain, for better feedback during the grasping or manipulation tasks of robots[7–11]. However, artificial sensors often lack the ability or perform insufficiently to perceive and recognize the real world upon touching the target objects[12,13]. By contrast, the human skin, especially the fingertip, not only feels and weighs but also helps identify the objects it touches[6,14,15]. The biological haptic perception involves the detection of both static pressure and high-frequency

vibrations: the slow adaptive (SA) receptors in the skin respond to the static pressure, and the fast adaptive (FA) receptors respond to subtly changed dynamic pressure—the rich frequency information provides a new dimension to understand the characteristics of the interaction and to identify the target objects[16–18]. A representative example is that humans can recognize braille alphabets or types of textiles by gently sliding a fingertip over those objects (textures with surface features). The identification of objects can in return feedback to manipulation—a human pinches an egg more carefully than holding a plastic ball with the same shape, size, and weight, while such discriminative dexterous manipulation is still challenging for robots due to their lack of touch-based object recognition. Recently, a few flexible sensor-based

[1]Department of Materials Science and Engineering, Southern University of Science and Technology, Shenzhen 518055, China. [2]School of Mechano-Electronic Engineering, Xidian University, Xi'an 710071, China. [3]Department of Computer Science and Engineering, Southern University of Science and Technology, Shenzhen 518055, China. [4]Department of Physics and TcSUH, University of Houston, Houston, TX 77204, USA. [5]These authors contributed equally: Ningning Bai, Yiheng Xue. ✉e-mail: guocf@sustech.edu.cn

artificial sensory systems inspired by biological sensory systems have been developed and shown the potential to realize subtle tactile sensation for machines[19–24].

A key challenge for the perception and recognition of fine surface features such as the texture or roughness of an object lies in the difficulty of achieving both high sensitivity and a rapid response-relaxation speed for both static pressure and vibration detection in flexible tactile sensors. Ultrahigh sensitivity is needed to allow a sensor to respond to weak stimuli during its interaction with tiny surface features, preferably down to a few microns; and a rapid response-relaxation speed is required for the sensor to resolve the characteristic spacings of surface features or to detect high-frequency and tiny vibrations. The existing work for fingertip sensing can hardly balance the two properties in a single sensor[25–27]. As a result, artificial sensory systems often use two sensors (together with two circuits that collect and process different types of signals), one for the detection of static pressure, and the other specifically for the detection of vibration. Furthermore, the correlation between the sensing performance and the recognition capability is still not fully understood. For example, a sensor with a wide frequency range is often pursued, while for discrimination a high temporal resolution (or frequency resolution) is at least equally important, but this has seldom been discussed.

In this work, we report a real-time and visual artificial sensory system of prosthetics based on a single flexible sensor, and we introduce spatiotemporal resolution as the criterion that determines the ability of a sensory system for texture recognition. The sensor utilizes tunable electric double layers (EDLs) that have a nanoscale charge separation for capacitive signals, giving rise to ultrahigh sensitivity up to 519 kPa$^{-1}$, in addition to a high spatial resolution down to 15 μm in spacing and 6 μm in height. Furthermore, the selection of low-viscosity ionic material together with the microstructural design allows the sensor to rapidly respond to high-frequency vibrations up to 400 Hz with a high frequency resolution of 0.02 Hz. The high spatiotemporal resolution allows a slip-sensor to discriminate tiny surface features with close spacings. We demonstrate that the real-time sensory system can be used for the classification of 20 different textiles with an average recognition accuracy of 98.6%, and the results can be real-time displayed in a visual interface. Such a system is expected to promote the sensing technologies of robotics and prosthetics, and is potentially useful for the sensory recovery of patients wearing artificial prostheses, haptics-based virtual reality, and consumer electronics.

## Results

### Concept of the artificial sensory system
In the human biological sensory system (Fig. 1a), a non-conductive potential change is generated when the skin perceives the outer world by the SA and FA cutaneous mechanoreceptors, and the signal is transmitted to the brain via the nervous system for further analysis and judgment (recognition)[6,17,28]. Our artificial sensory system mimics the function of the human sensory system by using only a sensor to realize both functions of the SA and FA receptors (Fig. 1b)−the sensor can respond to both static pressure and high-frequency vibrations during the physical interaction with textures or other objects. The signal with spatiotemporal information is further collected and transmitted using a circuit broad, and analyzed using machine learning with the recognition result being output in a visual user interface.

### Materials, structure, sensing properties, and mechanism for static and dynamic pressure detection of the slip-sensor
The sensor, which is called slip-sensor, consists of an artificial fingerprint made of polydimethylsiloxane (PDMS), an ionic gel layer of polyvinyl alcohol (PVA)-phosphoric acid (H$_3$PO$_4$, 8.3 wt.%), two flexible electrodes of a gold (Au) film on polyethylene terephthalate (PET), and a flat PDMS film for encapsulation (Fig. 2a). The artificial fingerprint consists of a set of concentric elliptical structures that have a triangular ridge with a height of 260 μm and an inter-structure spacing of 350 μm (Fig. 2b). Both the dimensions (Supplementary Fig. 1) and elastic modulus of the artificial fingerprint are close to that of the human fingerprints to effectively capture the vibrational stimuli during taction[18,29]. The PVA-H$_3$PO$_4$ gel has a graded, microstructured surface with two levels of structures: periodic domes with a diameter of 200 μm and a height of 55 μm, and finer protrusions that are densely distributed on the domes (Fig. 2c). The specific dimensions of the periodic domes or finer protrusions, including their diameters and heights, are determined by a trade-off between the fabrication resolution and the thickness of the device.

The graded microstructures of the ionic layer help improve the sensitivity and reduce the response-relaxation time of sensors. Here, the sensor exhibits an ultrahigh capacitance-to-pressure sensitivity up

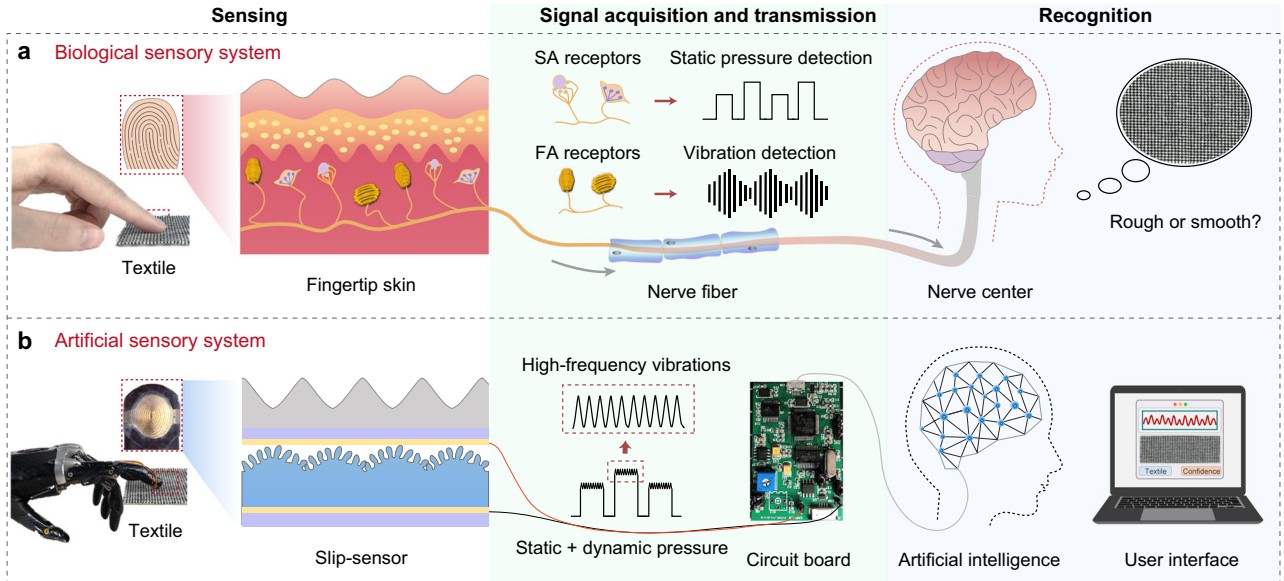

**Fig. 1 | A robotic sensory system mimicking the human sensory system for texture recognition. a** The biological sensory system of humans. **b** The artificial sensory system of this study, for which the sensor can detect both static and dynamic pressures.

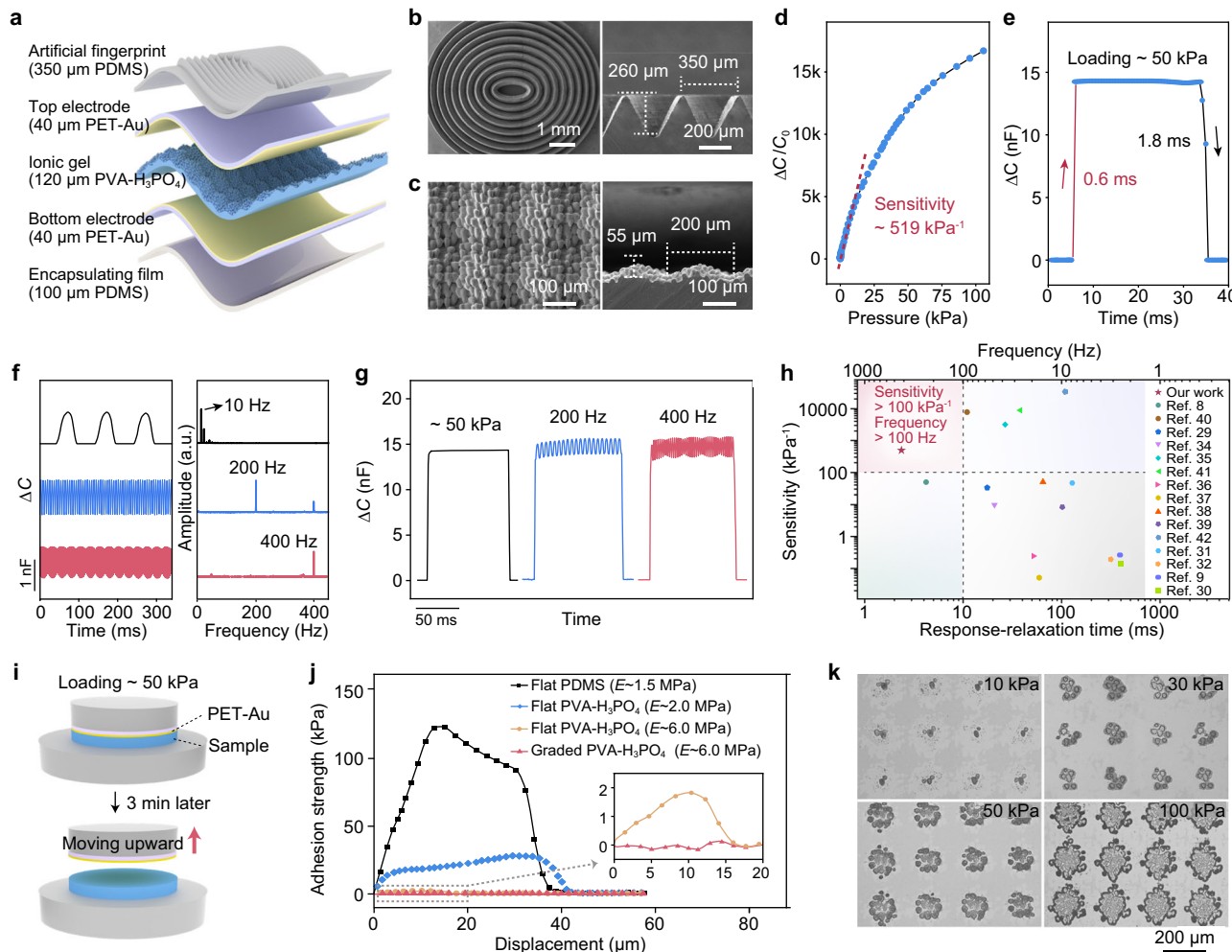

**Fig. 2 | Structure, sensing properties, and mechanism for static and dynamic pressure detection of the slip-sensor. a** Schematic diagram of the structure of the slip-sensor. **b** Scanning electron microscopy (SEM) images of the artificial fingerprint. **c** SEM images of the PVA-H$_3$PO$_4$ gel. **d** Normalized change of capacitance as a function of pressure over 100 kPa, showing a high sensitivity of 519 kPa$^{-1}$ at the low-pressure range. **e** Response-relaxation time (0.6 and 1.8 ms) of the slip-sensor. **f** Capacitive response of the sensor at vibration frequencies of 10, 200, and 400 Hz in the time-domain and the frequency spectra of the capacitance signals. **g** Static pressure (~50 kPa) detection and vibrations detection with frequencies of 200 and 400 Hz under a static base pressure of ~50 kPa. **h** Comparison between our sensor and existing capacitive sensors in terms of sensitivity, response-relaxation time, and corresponding frequency range. **i** Schematic diagram of adhesion strength test. **j** Adhesion strength between the PET-Au electrode and samples with different moduli. **k** Contact between the microstructured ionic gel and the electrode under different pressures.

to 519 kPa$^{-1}$ (Fig. 2d). The response maintains stable over cycling—no substantial signal drift is observed during 10,000 loading−unloading cycles with a peak pressure of 100 kPa (Supplementary Fig. 2). Furthermore, the sensor exhibits low hysteresis by loading a maximum pressure of 100 kPa and releasing (Supplementary Fig. 3).

The high sensitivity is attributed to the subtle change in microstructured EDL interface upon loading. Before applying pressure, the presence of an air gap prevents the contact between the electrode and the ionic gel, resulting in a low initial capacitance ($C_0$) of ~8 pF. When pressure is applied, the smaller protrusions of the ionic gel begin to contact with the electrode, and the signal increases sharply because of the increasing EDL capacitance. As the pressure further increases, the larger microdomes are involved in the contact, and the capacitance remains increasing (Supplementary Fig. 4). Therefore, such two-level microstructures increase the sensitivity and extend the working range of the sensor.

The response-relaxation speed is a crucial parameter that determines the capability of a sensor to detect high-frequency vibrations. Existing piezocapacitive sensors often exhibit a response-relaxation time of tens of milliseconds, corresponding to a vibration detection

limit on the level of 10 Hz[9,30−32]. Our sensor exhibits a rapid response time of 0.6 ms and a relaxation time of 1.8 ms upon loading (50 kPa) and unloading, adding up to a total response-relaxation time of ~2.4 ms (Fig. 2e). Such a rapid response-relaxation process enables the sensor to effectively respond to high-frequency vibrations up to 400 Hz, as shown in the time-dependent capacitance signals and the corresponding Fourier transform spectra (Fig. 2f). The response-relaxation time is almost two orders of magnitude shorter than that of existing capacitive sensors, and comparable to that of a rigid-soft hybrid sensor[33].

The slip-sensor can respond to a combined mode of static pressure and high-frequency vibrations, which is a common case that humans interact with the environment. Figure 2g shows the detected signals of three conditions: static pressure (50 kPa), 200 Hz vibrations (amplitude: ~5 kPa) superimposed to the static pressure, and 400 Hz vibrations superimposed to the static pressure—the vibrational signal is not interfered at such a high base pressure.

Our sensor is superior to existing capacitive sensors in terms of combined high sensitivity and rapid response-relaxation speed (or a wide frequency-response range, see Fig. 2h)[8,9,29−32,34−42]. Other types of sensors, say, piezoelectric and triboelectric sensors, although being

capable of capturing vibrational signals with a wider range, fail to detect static pressures[43,44].

We ascribe the rapid response-relaxation speed to both the materials selection and structure design. First, we select an ionic gel that has a high stiffness, which leads to a low viscosity and weak interfacial adhesion. This stems from that a highly crosslinked or stiff network has a shorter chain length and thus a lower viscosity. Here, the elastic modulus ($E$) of the gel is 5.5 MPa, much higher than that of the materials (PDMS, softer PVA-$H_3PO_4$, etc.) used in existing capacitive sensors[8,29,35,38]. The interfacial adhesion behavior between the electrode and the ionic gel determines the relaxation time of the sensor because the contact area is proportional to the capacitance value. We measured the adhesion strength versus the displacement upon loading 50 kPa, holding for 3 min, and release (Fig. 2i). The graded PVA-$H_3PO_4$ gel ($E \sim 5.5$ MPa) exhibits a negligible adhesion strength (close to 0 kPa). By contrast, three control samples, flat PVA-$H_3PO_4$ gel with a higher modulus ($E \sim 5.5$ MPa), flat PVA-$H_3PO_4$ gel with a lower modulus ($E \sim 2.0$ MPa), and flat PDMS ($E \sim 1.5$ MPa) (Supplementary Fig. 5), exhibit adhesion strengths of 1.8, 27, and 122 kPa, respectively (Fig. 2j). It is the low viscosity of the material and the low adhesion strengths of the interface that greatly reduce the rapid response and relaxation time. Second, the graded structure—microdomes with finer protrusions of the ionic gel, can lead to a small contact area between the electrode and the ionic gel (Fig. 2k). Such a small contact area further reduces the adhesion energy of the interface and increases the energy release rate of the system, thereby an increased response and relaxation speed. In addition, the ionic radii of hydrogen ions and inorganic anions in the PVA-$H_3PO_4$ gel enable faster ion migration, contributing to a rapid response-relaxation speed as well.

## Spatiotemporal resolution of the slip-sensor

The PDMS fingerprint on top of the sensor mimics the human fingerprint to capture the vibrational stimuli during interaction, and the fingerprint tip size determines the spatial resolution of the sensor. Without a fingerprint, although being highly sensitive, the sensor may not effectively interact with surface textures. We set the ridge tip width of the artificial fingerprint to be 13 µm (Fig. 3a), and tested the response of the slip-sensor when it slides on surface textures with fixed feature spacings ($l$) of 10, 15, and 50 µm (insets of Fig. 3b) at a sliding rate of 1.0 mm s$^{-1}$. Our results show that the slip-senor is capable of detecting features with a characteristic spacing larger than the fingertip size (e.g., features with a spacing of 50 or 15 µm), while failing to detect smaller features (10 µm) because the fingerprint tips cannot fill into the small gaps to interact with the structure (Fig. 3b). Such a size match can be verified in a further experiment: when the artificial fingerprint tip width increases to 25 µm (Supplementary Fig. 6a), the slip-sensor fails to detect surface structures with a spacing of 20 µm, but can respond to structures with a larger spacing (30 µm) (Supplementary Fig. 6b). In addition, signal magnitude increases with contact pressure due to the stronger interaction between the slip-sensor and the microstructure (Supplementary Fig. 7). Note that the height of the surface texture also affects signal amplitude (Fig. 3c). At a fixed spacing of 50 µm, the signal amplitude decreases with decreasing feature height from 50, to 30, and to 10 µm (Fig. 3d). We further verify that our slip-sensor could detect features with a spacing of 15 µm and a height of 6 µm (Supplementary Fig. 8).

The high spatial resolution (15 µm for spacing and 6 µm for height) allows the sensor to detect tiny surface features including the microscale fibers (tens of microns in diameter) of textiles. Furthermore, the spatial resolution of the slip-sensor is superior to that of the human fingertips. The human fingertip shows poor accuracies (with a 24.3% average accuracy, and accuracy ≤44% for all cases) together with a small Kappa coefficient (0.12, which indicates the lowest level of agreement with unreliable data[45]) in differentiating a nonstructured surface, and surface textures with feature sizes of 10, 15, 20, 30, 40, and

50 µm, based on our experimental results from five volunteer subjects (Supplementary Fig. 9). The result suggests that humans cannot resolve structures small than 50 µm after excluding random guessing.

Besides the high spatial resolution, the slip-sensor exhibits high temporal or frequency resolution as well. Figure 3e shows that the slip-sensor can effectively differentiate vibrations with close frequencies of 400.0, 400.1, and 400.2 Hz in the time domain, and such a difference can also be seen in Fig. 3f, where clear pulses corresponding to the frequencies are shown. The frequency resolution is determined to be ~0.02 Hz (or ~0.005% at 400 Hz), identified by the full width at half maximum (FWHM) of the peaks. Such a high frequency resolution allows the slip-sensor to identify surface textures with close feature spacings.

In an ideal case, the characteristic frequency $f$ of vibrations during the interaction between the slip-sensor and a surface texture is determined by the sliding rate $v$ of the sensor and the feature spacing $l$ of the texture:

$$f = v/l \tag{1}$$

This relation is verified in our experiment: when the slip-sensor slides on a blended textile (consisting of 56% polyester and 44% polyamide) with $l$ of ~275 µm (Fig. 3g) at different sliding rates (2, 20, and 100 mm s$^{-1}$), signals in the time-domain can be well detected (Fig. 3h), and the frequency-domain shows corresponding characteristic frequencies of 7, 73, and 365 Hz (Fig. 3i), respectively. Note that a base pressure of ~50 kPa is applied to the sensor upon sliding in case that intimate contact is lost. The moderate pressure also ensures the slip-sensor to capture high-frequency vibration at a high sliding rate of at least 100 mm s$^{-1}$, which is much faster than the typical sliding rate of human fingers. This allows the slip-sensor to work with high efficiency.

In addition to the feature spacing and height, the stiffness of the objects also affects the signal output of the slip-sensor. Unlike the characteristic frequency $f$ that is solely correlated with $l$, the signal amplitude depends on not only structure height but also the elastic modulus of the material. With the same geometry and size (Supplementary Fig. 10a), textures with a higher stiffness present a higher signal amplitude in the time-domain (Supplementary Fig. 10b) and frequency-domain (Supplementary Fig. 10c). Therefore, it is not valid to identify textures by simply considering signal amplitude or frequency, given that the relationship between the parameters of surface features and corresponding signal amplitude is undermined, together with the fact that many textiles have close structural spacings. Machine learning, as a powerful artificial intelligence technology that can use the measured data of sensors to predict outputs, is thus used for the recognition of textures in an artificial sensory system.

## Artificial sensory system for the recognition of textiles

We constructed an artificial sensory system based on the slip-sensor and leveraged machine-learning techniques for texture classification. A dataset was first built to train a model for the sensory system. The slip-sensor was set to slide on 20 textiles with different structures (Fig. 4a and Supplementary Fig. 11) and different materials including wool, cotton, and several blended fabrics (Supplementary Table 1) at a certain sliding rate. The structure periods of the textiles were measured based on microscopic observation, and it shows that the periods of most textiles are close (Fig. 4b)—some of the differences are within 2.5% (e.g., 449 µm for textile no. 9 and 439 µm for no. 19). Correspondingly, the characteristic frequencies are also close at a given sliding rate (Fig. 4c, which shows the results at sliding rates of 2 and 40 mm s$^{-1}$), while our sensor can well discriminate such tiny differences in both spatial and frequency domains because of its adequately high spatiotemporal resolution. A simple comparison can illustrate the capability of our sensor to distinguish such textiles: it has a high

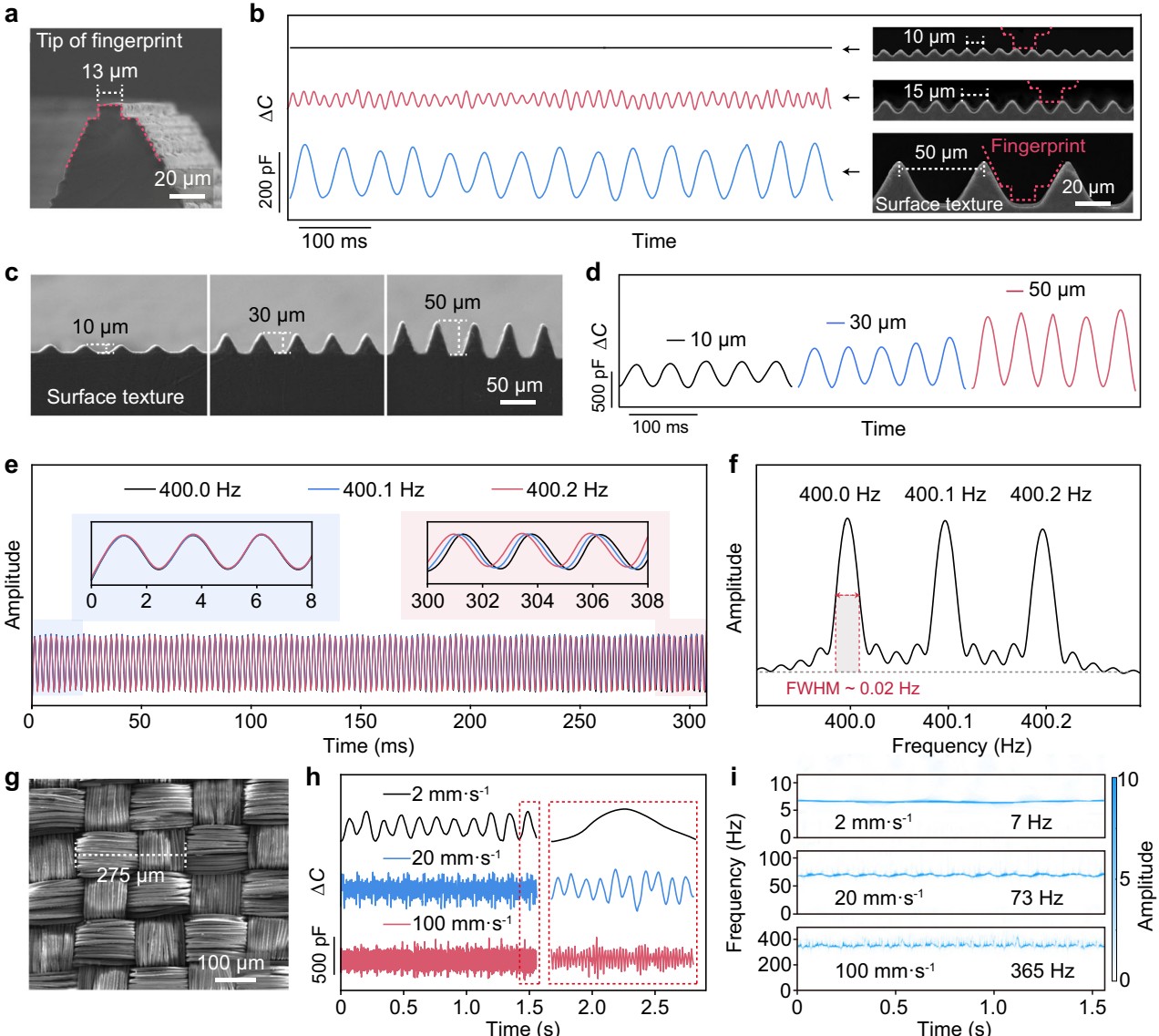

**Fig. 3 | Spatiotemporal resolution of the slip-sensor. a** SEM image of the artificial fingerprint with a tip width of 13 µm. **b** Signals generated by sliding the slip-sensor on surface textures with feature spacings of 10, 15, and 50 µm at a sliding rate of 1 mm s⁻¹. Insets show the SEM images of the textures. **c** SEM image of textured structures that have ridges with three different heights (10, 30, and 50 µm). **d** Signal generated by the interaction of a slip-sensor with the fine structure in **c**.

**e** Vibrational signals with frequencies of 400.0, 400.1, and 400.2 Hz detected using the slip-sensor. **f** Frequency spectra of the signals in **e**. **g** SEM image of a textile with a feature spacing of ~275 µm. **h** Collected signals by sliding the sensor on the textile at three different sliding rates (2, 20, and 100 mm s⁻¹). **i** Frequency-domain diagrams of the signals in **h**.

frequency-resolution of 0.005%, while the smallest frequency difference between the textiles is close to 2.5%.

We use a readout circuit board to collect the instantaneous signals generated when the sensor slides across the textiles. The circuit board consists of five parts: a power supply module, a microcontroller module (STM32) serving as the central processing unit to process data and make decision, an input/output interface module for communication with external devices, a signal processing module responsible for conditioning internal signals, and a 24-bit analog-to-digital conversion (ADC) module for sampling signals (Supplementary Fig. 12).

Figure 4d shows the distinguishable signals in the time-domain collected at a sliding rate of 2 mm s⁻¹ for the 20 textiles, and the distinctive frequency-domain features of the signals are calculated using wavelet transformation (Supplementary Fig. 13). Considering that the complex factors (height, spacing, stiffness, etc.) of the textiles all affect the output signal, we use machine learning to classify these complex

features. We use T-distributed stochastic neighbor embedding (t-SNE) to project higher-dimensional data into a two-dimensional (2D) space to visualize the data while preserving their global and local structure. Figure 4e presents distinguished clusters of different sets of data collected from the 20 textiles, showing that the datapoints of the objects can be well visualized and distinguished in a 2D space.

We adopt a Bagging ensemble learning approach to solve the classification problem, which improves the generalization capability of the model and the overall classification performance. The classification models (or classifiers) employed in the ensemble include K-nearest neighbors, random forests, logistic regression algorithms, and decision trees. In order to accurately distinguish between different textures, we used the Tsfresh tool—a Python package to extract hundreds of time-series features, such as statistical, frequency-domain, autoregressive, wavelet transform, and time-domain features (Supplementary Table 2). There are 20 categories and dozens of features in

our dataset, and Fig. 4f shows four categories and two features for simplified illustration. Each category has 100 sets of data, which were divided into five blocks. We selected one block at a time as the testing

set and the rest as the training data, and iterated the prediction results in the testing set several times. By combining the predictions of each base classifier through voting, more accurate and robust overall

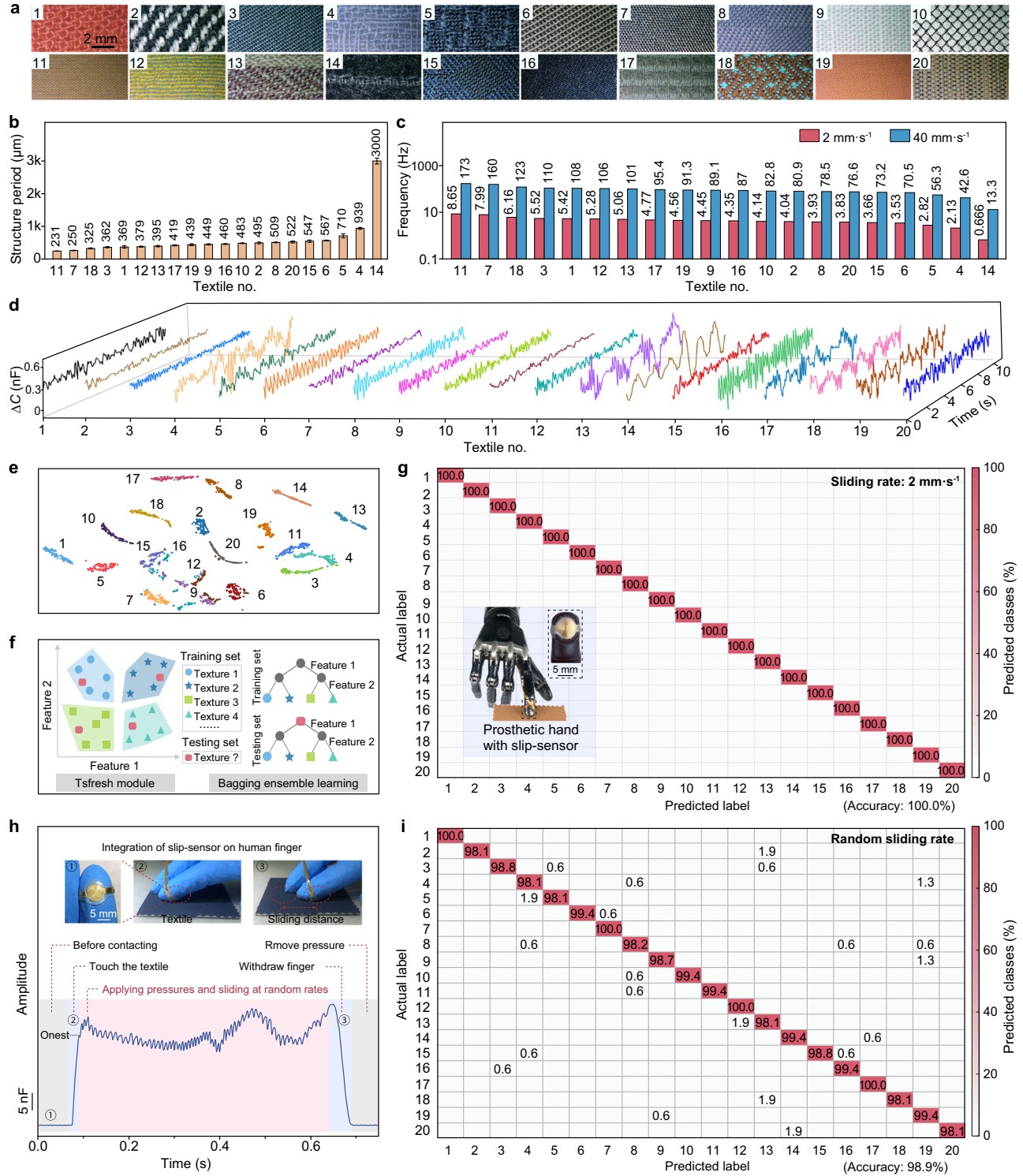

**Fig. 4 | Recognition of textiles using the sensory system. a** Digital photos of 20 different textiles. **b** Structure periods of the 20 textiles. Error bars represent the standard deviation of the structure periods. **c** Characteristic frequencies of the 20 textiles at sliding speeds of 2 mm s⁻¹ and 40 mm s⁻¹. **d** Time-domain signals of 20 textiles sensed using the slip-sensor at a sliding rate of 2 mm s⁻¹. **e** T-SNE visualization of the data set collected from the 20 textiles. **f** A schematic of the feature extraction and signal classification. **g** Confusion matrix for the recognition of the 20

different textiles at a sliding rate of 2 mm s⁻¹ using the artificial sensory system integrated with the slip-sensor on a prosthetic hand. A recognition accuracy of 100.0% is achieved. The inset shows the prosthetic hand integrated with a slip-sensor. **h** Time-domain signals if the sensor integrated on a human finger by sliding on textile no. 16 at random sliding rates and contact pressures. **i** Confusion matrix for the recognition of the 20 different textiles at random sliding rates and contact pressures, showing an average recognition accuracy of 98.9%.

classification results can be obtained. The output confusion matrix finally confirms an accuracy of 100.0% for textile recognition (Fig. 4g). The high recognition accuracy can be attributed to two aspects. First, the slip-sensor can sensitively capture the small differences between different textiles. Second, the tactile dataset for each textile is consistent at a fixed sliding rate. We extracted a multitude of features using the Python package tsfresh that helps achieve a high classification accuracy. The high recognition accuracy can be maintained at higher sliding rates for enhanced working efficiency. At a higher sliding rate of 40 mm s$^{-1}$, the system can still achieve a high recognition accuracy of 99.5% (Supplementary Fig. 14). The human fingers touch objects at variable sliding rates, we thus conducted texture recognition at a random sliding rate. We fixated the slip-sensor on the index finger of a human subject and unconsciously slid the sensor over the textiles with unknown contact pressure and sliding rate. Here, we collected signals for the whole interaction between the sensor and the textile: before contacting, finger touching, finger sliding over the textile, and finger withdrawing (Fig. 4h and Supplementary Fig. 15).

Considering the potential consistency within the sliding habits of the same subject, we collected data on random sliding rates from three different subjects. Each category involved random touches from three individuals in a 2:1:1 ratio, resulting in a total of 400 sets of data for each category and a total of 8000 sets for the 20 textile types, with 40% of the data reserved for testing. We used an inertial measurement unit (IMU) to record the acceleration during the sliding process. The evidently chaotic acceleration in the x-y plane confirms that the sliding rates of an individual continuously change during sliding (Supplementary Figs. 16–18). An average recognition accuracy of 98.9% was achieved, revealing the high robustness and reliability of our sensory system in texture recognition (Fig. 4i). We ascribe the higher recognition accuracy of our artificial sensory system to the high spatiotemporal resolution: our sensor can respond to tiny surface features and differentiate yarns even if they have close spacings. Note that the signal for the onset of the slip (Fig. 4h) may reflect extra features (such as friction coefficient) of the texture to further improve the classification accuracy.

We further constructed a portable sensory system with a real-time and visual user interface displayed on a PC for intuitive classification of surface textures. This system consists of a slip-sensor integrated into a fingertip of prosthetic or human hand, a circuit board for collecting sensing information of textures and sending real-time data to a PC via USB wired transmission, the aforementioned machine learning method for classifying the textures, and a user interface for visualizing the output results (Fig. 5a).

In the real-time sensory system, all 2000 data sets collected at a sliding rate of 2 mm s$^{-1}$, and all 8000 data sets collected at random rates were used for training, while the test sets that consisted of independent data were collected by the circuit board in real-time. By analyzing the real-time data, features were extracted and classification models were applied to make immediate predictions or recognitions. Real-time recognition enables rapid decision-making and faster feedback based on streaming data, making it suitable for time-sensitive applications or scenarios that require immediate response. Our experimental results showed that the inference time was below 20 ms, validating the real-time feasibility of our sensing system. With the machine learn-based classifier, we can identify the textiles in the signals collected in real-time and display the confidence of the recognition as well as the microscopic morphology of the textiles identified on a real-time visual user interface. Figure 5b illustrates an implementation of the real-time sensing systems, showing the high confidence when a prosthetic hand with an integrated slip-sensor touches textiles at a sliding rate of 2 mm s$^{-1}$. Of particular note is that the real-time system shows an average accuracy of 100.0% for these 20 textiles (Supplementary Movie 1). When the sensor slides over the same textile at variable rates, the confidence slightly decreases due to the diversity and complexity of the data (Fig. 5c). However, the sensory system

consistently achieves an average recognition accuracy of 98.6% for these 20 textiles (Supplementary Movie 2).

Existing work has reported several sensory systems for texture or material recognition using two types of sensors (e.g., piezoresistive and piezoelectric sensors, or piezoresistive and triboelectric sensors) to simulate SA and FA receptors[25,26,46–49]. These sensory systems, however, require two sensors integrated together with two sets of data acquisition systems (Supplementary Table 3). In contrast to these sensory systems, our sensory system can achieve a high recognition accuracy using a single sensor, while the system is simplified and robust.

## Discussion

The fine fingerprint plays a key role in allowing the sensor to fully interact with the fine features of textures, even at high sliding rates. Without the fingerprint, the recognition accuracy drops to only 54.5% at a sliding rate of 2 mm s$^{-1}$ (Supplementary Fig. 19). Furthermore, the slip-sensor exhibits a high signal-to-noise ratio of 86.79 dB and a high effective number of bits of 14.12 bits (Supplementary Fig. 20). These characteristics ensure that the sensor can precisely capture subtle tactile signals and deliver high-quality output in texture recognition.

Overall, our artificial sensory system has a higher accuracy in the differentiation of fine textures. Such an artificial sensory system is not only potentially useful in robotics, but also expected to be applied in healthcare and consumer electronics by helping humans achieve enhanced haptic functions, and by offering new technologies for metaverse.

## Methods

### Fabrication of the slip-sensor

The fabrication of the slip-sensor included the preparation of the artificial fingerprint, the flexible electrodes, the ionic gel, and the encapsulation of the device (Supplementary Fig. 21).

### Fabrication of the artificial fingerprint

A reverse template of fingerprints that have a similar aspect ratio and a depth-width ratio to the human fingerprint was constructed by 3D modeling, and printed out using resin by high-precision 3D printing (NanoArch S130, BMF Precision Tech, Inc.). All other resin microstructures were also prepared using high-precision 3D printing. PDMS base and curing agent (Sylgard 184, Dow Corning Co., Ltd.) with a mass ratio of 5:1 were cast on the surface of the fingerprint mold. After curing at 80 °C for 30 min, the PDMS fingerprint (thickness: ~350 μm) was peeled off from the mold.

### Preparation of the ionic gel

A resin mold with an inverse-graded microdome structure was 3D-printed and used as a mold for preparing the microstructured ionic film. Two grams of PVA (Mw ~145,000, Aladdin Industrial) was dissolved in 20 g deionized water and stirred at 90 °C for 2 h. Next, the PVA solution was cooled to 50 °C, and 1.5 mL H$_3$PO$_4$ (AR, ≥85%, Shanghai Macklin Biochemical Co., Ltd.) was added and stirred continuously for 1 h. The PVA-H$_3$PO$_4$ mixture was then casted onto the surface of the resin mold and cured for 24 h at a temperature of 24 °C and 43% humidity. Finally, the PVA-H$_3$PO$_4$ films (thickness ~120 μm) were removed from the mold and cut into circles with a diameter of 7 mm for later use.

### Preparation of flexible electrodes and sensor encapsulation

A layer of 100-nm-thick Au film was deposited on 40-μm-thick PET (HD, DuPont) using ion sputtering (MC1000, Hitachi High-Tech), serving as the electrodes for the sensor. The PET-Au film was then cut into circles with a diameter of 7 mm for later use. The slip-sensor consisted of five layers from top to bottom: artificial PDMS fingerprint, a PET-Au top electrode, a PVA-H$_3$PO$_4$ layer with graded microdomes as the

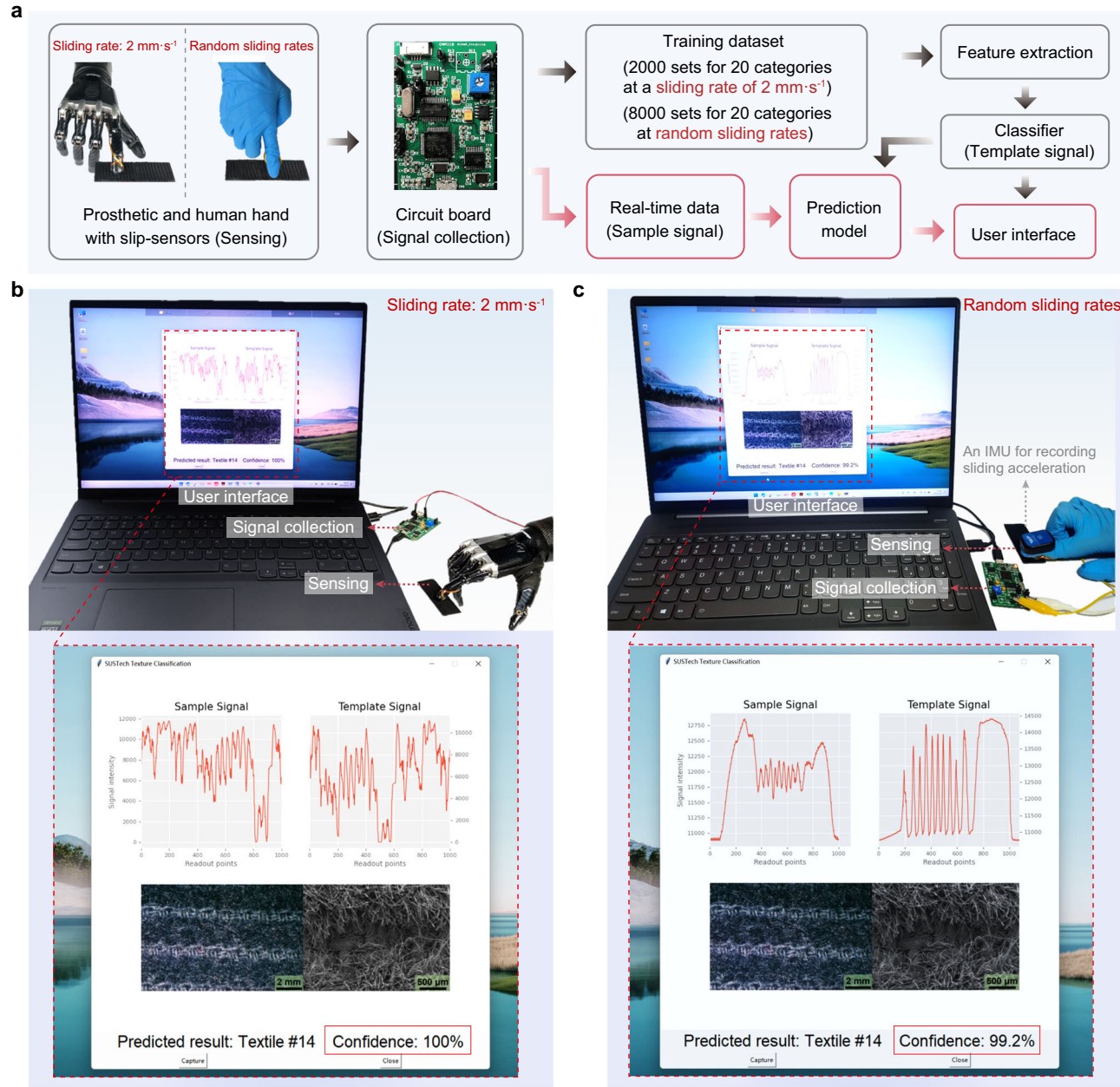

**Fig. 5 | A portable and real-time sensory system with a visual user interface.** **a** Structure of the real-time sensory system. **b** Demonstration of the real-time visual user interface for the recognition of textiles using a prosthetic hand integrated with a slip-sensor working at a sliding rate of 2 mm s$^{-1}$. **c** Demonstration of the real-time visual user interface for which the sensor works at random sliding rates.

active layer, a PET-Au bottom electrode, and a flat PDMS membrane (thickness: 100 μm) as a bottom encapsulation material. The PDMS fingerprint layer and the flat PDMS membrane were treated by plasma and bonded to the bare side of the PET layer. Finally, the PDMS fingerprint and the flat PDMS film were bonded with silicone adhesive (Sil-Poxy, Smooth-On, Inc.).

## Characterization and measurements

The morphology of the PDMS fingerprints, the ionic film, and the textiles were characterized by field-emission scanning electron microscopy (TESCAN MIRA3). A computer-controlled mechanical testing machine (XLD-20E, Jingkong Mechanical Testing Co., Ltd) was used to load the external pressure. The signals of the sensor were recorded using either a high-speed LCR meter (TH2840B, Tonghui) or a digital circuit board. For the consideration of high sampling frequency, all high-frequency vibration signals, including response-

relation speed, were collected using an LCR at a testing frequency of 10 kHz with a sampling frequency was ~1600 Hz. The circuit board was used in the artificial sensory system to acquire digital signals with a sampling frequency of 1000 Hz.

The sensitivity was defined as $S = \delta(\Delta C/C_0)/\delta P$, where $P$ represents the applied pressure, and $\Delta C$ was the difference between the measured capacitance $C$ and the initial capacitance $C_0$ (~8 pF). Increased dynamic pressures up to 100 kPa were applied to the sensor, and the corresponding peak capacitance value for each pressure was recorded using the LCR meter at the test frequency of 1000 Hz. The slope in the $\Delta C/C_0$-$P$ curve represented the sensitivity value of the sensor. The response-relaxation time was tested by rapidly applying and withdrawing a pressure of ~50 kPa to the sensor using manual pulling on a flat metal post with a diameter of 3 mm. The time for the rising edge represented the response time, and the time for the descending edge was the relaxation time.

A vibration generator (Model BL-ZDQ-2185, Hangzhou Peilin Instrument Co. Ltd.) was used to apply constant frequency vibrations to the sensor at a pressure of ~10 kPa, and the corresponding vibration response signals of the sensor were collected using the LCR meter. Electrochemical etching was adopted to characterize the change in contact area between the microstructured ionic gel and the electrode (Fig. 2k). A PET-Au electrode was used as the bottom electrode, and a copper film which was deposited on silicon was used as the top electrode, with the microstructured ionic gel sandwiched in between as the electrolyte. The etching was performed for 10 s at fixed pressures of 10, 30, 50, and 100 kPa at a bias of 1.5 V (CORRTEST, CS350).

The textiles were purchased from local markets. The structure periods of the textiles were measured by microscopic observation (Supplementary Fig. 11), three times for each textile in order to calculate the corresponding average structure period and error bar (Fig. 4b). A mechanical test platform, consisting of a tensile machine and a lifting platform (LZ80-2, OMTOOLS), was used for texture recognition test at fixed sliding rates. A polymethyl methacrylate rod bonded with a slip-sensor was fixed to the tensile machine, and the objects under test were fixed to the surface of the lifting platform. The pressure applied on the slip-sensor was controlled by adjusting the height of the lifting platform. The sensor slid across the surface of each object driven by the tensile machine to interact with its surface and corresponding capacitance signal was recorded. The data for the 20 textiles were collected at a sliding distance of 40 mm and a sliding rate of 2 mm s$^{-1}$, with a total of 2000 data sets. A static pressure of ~50 kPa was applied to the sensor before sliding. The texture recognition test at random sliding rates was conducted by fixing the slip-sensor on the subject's index finger and randomly sliding the sensor onto the textile with estimated sliding rates of 0–30 mm s$^{-1}$. The details about the recognition of microstructure by human subjects were provided in the Supplementary Information.

### Design of digital circuits

The design principle of the circuit is as follows. First, a 12-bit digital-to-analog converter inside the STM32 microcontroller was programmed to generate a stable sine wave signal as the excitation source for the measurement. This excitation signal was applied to the capacitor to be measured, causing a current to flow through the capacitor. During this process, the capacitor exhibited a fixed resistive characteristic known as capacitive reactance, which was proportional to the capacitance value. Next, a reactance-to-voltage conversion circuit was used to transform the capacitive reactance into a voltage signal that is proportional to the capacitance value. Subsequently, the output voltage signal was processed through a low-pass filter to obtain a DC voltage. Finally, this DC voltage was sampled using a 24-bit ADC, and the STM32 microcontroller calculated the capacitance value.

The circuit board had dimensions of 5.5 cm in length and 3.5 cm in width (inset in Supplementary Fig. 12), which were determined based on the functional requirements and component quantity.

### Construction of the real-time sensory system with a visual user interface

In the real-time artificial sensory system with a visual interface, the signals were collected using a digital circuit board. Python package tsfresh was used to extract the features of textures, and random forest algorithms were used for classification and analysis. A user interface was designed and displayed on a PC to provide real-time feedback on the predicted signal waveform, the real sampled signal waveform, and the microscopic image of the textile, together with the predicted confidence value. The experiments were conducted on a laptop with 64 GB of RAM, and the preprocessing and classification times were consistently <20 ms per data point.

### Reporting summary

Further information on research design is available in the Nature Portfolio Reporting Summary linked to this article.

## Data availability

Data generated in this study are provided in the Main Text and the Supplementary Information. Additional data are available from the corresponding author upon request.

## Code availability

The MATLAB code for wavelet transformation and the Python codes supporting the portable and real-time sensory system for texture recognition are openly available on GitHub at https://github.com/Billy1203/SUSTech-texture-recognition.

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

## Acknowledgements

The work was supported by the "National Natural Science Foundation of China" (No. T2225017, 52073138), the "Science Technology and Innovation Committee of Shenzhen Municipality" (No. JCYJ20210324120202007), and the "Guangdong Provincial Key Laboratory Program" (No. 2021B1212040001).

## Author contributions

C.F.G. conceived the idea and directed the study. N.B. conceived the idea and conducted the majority of experiments. C.F.G. and N.B. wrote the manuscript. Y.X. performed the machine learning and designed the visual user interface. S.C. and J.Z. designed digital circuits for data acquisition. L.S. joined in the test of texture recognition. J.S. wrote the code of MATLAB. N.B., Y.Z., X.H., Y.C., and K.H. carried out the recognition of microstructures. W.W. and Y.L. polished the manuscript. All authors approved the final version of the manuscript.

## Competing interests

The authors declare no competing interests.
