## [Peer Review File · Nature Communications]

REVIEWER COMMENTS

Reviewer #1 (Remarks to the Author):

The manuscript by Bai et al. reports an artificial sensory system mimicking the biological sensory system for texture recognition. This system uses a specially designed iontronic sensor that can detect both static pressure and dynamic stimuli up to 400 Hz, which can otherwise be measured using two different sensors. The authors introduce the concept “spatiotemporal resolution” for high-performance recognition, and they show that the iontronic sensor can exhibit a superhigh spatial resolution of around 10 microns, and a high frequency resolution of at least 0.1 Hz at a frequency of 400 Hz. Based on the high sensing performance of the sensor, the authors build a portable sensory system, and integrate the system with a prosthetic hand for real-time texture recognition, with the results to be displayed in a user’s visual interface. The system achieves a high recognition accuracy of 98.5% in classifying 20 commercial textiles, and an accuracy of ~97% in classifying fruits and other stuff. The authors show that the results are much higher than the classification accuracies by human subjects. Overall, this manuscript is of high novelty and will be of interest to audiences from many fields. I therefore strongly recommend publication of the manuscript in Nature Communications.

A few minor points for the authors to address:

1. The authors should provide the initial value of capacitance (C_0 , capacitance before applying any load). This is because the sensitivity value relies strongly on C_0 .
2. The Young’s moduli of the ionic materials (7.5 MPa) and other materials are questionable. Young’s modulus should be the slope of the stress-strain curve starting from the origin. Please correct the problem.
3. I suggest that the authors provide the data for the onset the slip, that is, the signal from the static friction to kinetic friction. The data might provide audiences with additional information (such as friction coefficient) for object recognition, although it is out of the scope of this work.
4. It seems to me that the spatial resolution of the sensor is determined by the size of the fingerprint tip. But why not further reduce this size to achieve a higher spatial resolution? Is that technically unavailable?
5. The frequency resolution is determined to be 0.1 Hz. The true frequency-resolution should be even higher because the authors tried only frequency difference of 0.1 Hz. Sometimes people use the FWHM of the peaks as the resolution.
6. There is a problem in the format of references (Ref. 19 and 20) in page 5. Please correct it.
7. In figure 1a, I suggest that the authors change the expression “rough or fine” to “rough or smooth”.

Reviewer #2 (Remarks to the Author):

The authors propose a artificial sensory system that can respond to both static and dynamic stimuli with high spatial resolution and high frequency resolution. By leveraging machine learning analytics, high-precision discrimination of textile surface textures have been

demonstrated for robotic applications. Although the sensor in this paper shows a great advantage in sensitivity, response relaxation time, and corresponding frequency range compared to other capacitive sensors, the novelty of its system level application is not particularly obvious compared to current state-of-art works, which needs to be strengthened. Thus, I suggest major revision with the following comments properly considered:

1. For the sensor structure, the periodic domes are designed with a diameter of 200 μm and a height of 55 μm . The choice of this parameter or the related optimization data/discussion are suggested to be mentioned in the text.
2. Typos and errors, e.g., “diamater” in “...with a diamater of 200 μm and a height of 55 μm ,”. The authors need to check through the main text again carefully.
3. For the spatiotemporal resolution test of the slip sensor, the authors fixed the sliding rate to 1.0 mm s⁻¹. However, the contact force/pressure may also affect signal amplitude during sliding, which needs to be mentioned or discussed in the text.
4. In Fig. 4, the authors demonstrated that the developed artificial sensory system can achieve high-accuracy textile recognition with a given sliding rate. However, in practical applications, the sliding rate is not always well maintained. What is the performance of this perception system at a relatively random sliding rate, e.g., to simulate the relatively random touch done by a human hand?
5. In Fig. 2, the authors demonstrated that the proposed sensor achieved the static and pressure detection simultaneously, and mimic the SA and FA of biological sensory system. However, for the textile recognition shown in Fig. 4 and Fig. 5, the authors only use the dynamic signal as the input of perception model. Actually, some works have proved that the fusion of static and dynamic features can obtain better texture recognition performance, i.e., Nano letters 19.5 (2019): 3305-3312. Such data is suggested to be added to make the story of the whole article seem more complete.
6. Many recent works have shown the possibility of simulating SA and FA for texture recognition under flexible sensory platform for HMI and robotic applications. A comparison table is needed to compare this work with other state-of-art works to further highlight the novelty of this paper.
7. Some recent interesting works can be referred in introduction to broaden the view of readers, e.g., Science Advances 8.31 (2022): eabq2521; ACS Nano, vol. 17, no.5, 4985-4998, 2023; ACS Nano, vol. 17, no.2, 1355-1371, 2023; Adv. Energy Mat., vol. 13, no.1, 2203040, 2023; Nat. Commun., vol. 13, 5224, 2022.; Applied Physics Rev, vol. 7, no. 3, 031305, 2020.

Reviewer #3 (Remarks to the Author):

The authors present a novel sensor based on tunable electric double layers (EDLs) that have a nanoscale charge separation which translates mechanical stimuli into capacitive signals. The sensor responds to static and dynamic stimuli and shows impressive performance i.e. sensitivity up to 519 kPa⁻¹, spatial resolution down to ~15 μm in spacing and 6 μm in height, frequency vibrations measurement sensitivity up to 400 Hz, frequency-resolution of 0.1 Hz,

total response-relaxation time of ~ 2.4 ms. The fingerprint has a bio-mimetic structure and geometry. The authors presents results on the recognition of 20 different commercial textiles with high accuracy and a portable system with a robotic hand which integrates the sensor, a PCB for interfacing with a PC and for real-time graphic interface for surface classification.

Strong points:

Novel iontronic sensor which measures both static and dynamic pressure/mechanical stimuli with low (for a capacitive sensor) response-relaxation time, high frequency bandwidth and frequency resolution.

Weak points:

Missing relevant information on the experimental setup (see other comments).

The machine learning approach looks overestimated: Fig. 4.e shows that, with a proper transformation, different clusters are almost separated, a simpler algorithm (e.g. SVM) could perform the task. The authors should compare the classification results with a reference algorithm in the state of the art.

The comparison with the human capabilities in textile recognition by touch exploration is not meaningful, I expect that an artificial system overcomes human performance in many different ways. In the proposed results, the artificial system applies the stimulus in a very controlled way which oversimplifies the task. This part of the paper is not meaningful and should be removed.

Presentation in some points is obscure, more details and a clear presentation must be supplied in many parts.

In the last part, the paper presents a touch-based object recognition experiment. This part should be revised as the task performed is the recognition of surface objects by sliding the sensor on the surface itself.

The read-out electronic circuit of the portable system must be clearly presented.

The claim in the Introduction “the selection of low-viscosity ionic material together with the microstructural design allow the sensor to rapidly respond to high-frequency vibrations” is not sufficiently motivated/demonstrated in the paper. I see two different points here: 1) the fingerprint arrangement which is biomimetic and improves the spatial resolution; 2) the ionic-material as insulating layer of the sensing capacitor which allows to achieve high frequency. The authors should better explain these two points.

Which are the advantages of using ionic-material with respect to other materials?

The advantages of the “microstructured surface with two-levels of structures” (see page 5) should be better discussed and demonstrated. The measurement of the static capacitance versus normal mechanical stimulus should be reported. The way the microstructured surface affects the capacitance value upon loading should be introduced.

The authors must explain the reason why “the hierachical microstructures of the ionic layer help improve the sensitivity and reduce the response-relaxation time of sensors.” Page 6, top. The section structure with the thickness of the sensor various layers must be reported.

Due to the total response-relaxation time of ~ 2.4 which is very impressive for a capacitive sensor, the corresponding maximum frequency that can be measured by the sensor is 416 Hz which overcomes capacitive sensors at the state of the art.

The authors should explain how the measurement setup of Fig. 2.d, if the measurements was

static or dynamic, they should report the values of ϵC and C_0 , it is not clear to me how they computed the sensitivity.

Fig. S2 should report the input stimulus versus time; 1000 cycles is not a relevant value to evaluate signal drift. I would increment the number of cycles by one order of magnitude at least.

The authors should make experiments to evaluate the hysteretic behavior of the sensor.

Which is the sensor output sampling rate?

What are piezocapacitive sensors (page 6, top)?

The thickness of the Ionic gel is 120 microns, the Ionic gel implements the insulating layer of the capacitor, which is the dielectric constant of the material? The capacitance value is in the order of nF (see Fig. 2.e), which is the capacitance value per unit area? When the mechanical stimulus is applied, the thickness of the ionic gel decreases, and the capacitance increases, which is the maximum capacitance value which has been measured during experiments? Which is the corresponding ionic gel thickness?

In Fig. 2.f, the amplitude value must be reported on the y-axis. Which is the input stimulus amplitude? Which is the measurement setup?

The sentence at page 7 “The interfacial adhesion behavior between the electrode and the ionic gel determines the relaxation time of the sensor” should be better explained also with experimental evidence.

The motivations of the frequency resolution of the sensor of 0.1 Hz should be clearly discussed.

The electronic interface circuit for capacitive variation measurement must be presented. Fig. S9 does not help in understand the read-out circuit.

Miniaturization of the read-out PCB should be discussed.

The organization of the data set e.g. time duration of each sample, sliding speed of each sample, static pressure of each sample, must be provided by the authors.

In Fig. 4.g are samples of different sliding velocities collected? Or only one sliding velocity has been used? It looks that experiments have been conducted with a constant sliding velocity. What happens with a dataset collecting samples with different sliding velocities?

Sentence in the Introduction: “However, artificial sensors often lack the ability or perform insufficiently to perceive, recognize, and explore the real world upon touching the target objects.”, exploration is an active action which is not implemented by sensors, I suggest removing it.

The information about experimental setup and dataset structure is not sufficient; the authors should supply all the features of samples used to build the dataset.

The authors should evaluate results with a dataset of samples collected at different velocities and different static pressure. I believe that if the sliding velocities and static pressure are constant the operation of the Machine Learning algorithm is over simplified as shown in 4.e where clusters of different textiles are clearly identified and do not overlap. This is in my opinion the very high classification accuracy that is achieved. Humans cannot precisely control the sliding velocity as such their performance are not that high if compared with those of a machine learning algorithm with data collected with precisely controlled operating conditions. The inference time, i.e. the time needed for output the classification result should be reported in the paper.

The training and test sets, the related operating conditions (see Fig. 5), should be clearly

reported in the paper. Which are the differences with respect to what reported in Fig. 4? Which is the meaning of the following sentence “all the 2000 sets of data were used as the training set, and we extract features for each textile to further establish a classification model”, which features are extracted? Which classification model has been used?

It is claimed that the system can identify macroscale objects (page 13), my understanding is that the sensor can classify different surfaces by sliding the sensorized finger on the object surface. The operating conditions of such experiment must be clearly reported. Discussion on results is confuse, the assessment is not convincing.

The authors must compare their results with the ones of: Zhang, J., Yao, H., Mo, J. et al. Finger-inspired rigid-soft hybrid tactile sensor with superior sensitivity at high frequency. *Nat Commun* 13, 5076 (2022). <https://doi.org/10.1038/s41467-022-32827-7>

Response to reviewers for manuscript NCOMMS-23-15409A-Z

Reviewer #1 (Remarks to the author):

The manuscript by Bai et al. reports an artificial sensory system mimicking the biological sensory system for texture recognition. This system uses a specially designed iontronic sensor that can detect both static pressure and dynamic stimuli up to 400 Hz, which can otherwise be measured using two different sensors. The authors introduce the concept “spatiotemporal resolution” for high-performance recognition, and they show that the iontronic sensor can exhibit a superhigh spatial resolution of around 10 microns, and a high frequency resolution of at least 0.1 Hz at a frequency of 400 Hz. Based on the high sensing performance of the sensor, the authors build a portable sensory system, and integrate the system with a prosthetic hand for real-time texture recognition, with the results to be displayed in a user’s visual interface. The system achieves a high recognition accuracy of 98.5% in classifying 20 commercial textiles, and an accuracy of ~97% in classifying fruits and other stuff. The authors show that the results are much higher than the classification accuracies by human subjects. Overall, this manuscript is of high novelty and will be of interest to audiences from many fields. I therefore strongly recommend publication of the manuscript in Nature Communications.

Response: We thank the reviewer for the kind and positive comments on our work.

A few minor points for the authors to address:

Q1. The authors should provide the initial value of capacitance (C_0 , capacitance before applying any load). This is because the sensitivity value relies strongly on C_0 .

Response: The value of the initial capacitance C_0 is ~ 8 pF and the information has been added in the “Methods” section in the revised manuscript.

Modification: Line 13, Page 6

“.....a low initial capacitance (C_0) of ~8 pF.”

Q2. The Young’s moduli of the ionic materials (7.5 MPa) and other materials are questionable. Young’s modulus should be the slope of the stress-strain curve starting from the origin. Please correct the problem.

Response: Thanks to the reviewer for correcting the method to calculate Young’s modulus. The corrected Young’s modulus is ~5.5 MPa, and the information has been revised in **Supplementary Fig. 5** and corresponding main text (Line 22, Page 7).

Q3. I suggest that the authors provide the data for the onset the slip, that is, the signal from the static friction to kinetic friction. The data might provide audiences with additional information (such as friction coefficient) for object recognition, although it is out of the scope of this work.

Response: The signal for the onset the slip has been provided (see revised **Fig. 4h**). The signal amplitude decreases during the transition from static friction to dynamic friction. Such data might provide extra information for texture recognition.

Modification: Line 7-8, Page 13

“.....Note that the signal for the onset of the slip (Fig. 4h) may reflect extra features (such as friction coefficient) of the texture to further improve the classification accuracy.”

Q4. It seems to me that the spatial resolution of the sensor is determined by the size of the fingerprint tip. But why not further reduce this size to achieve a higher spatial resolution? Is that technically unavailable?

Response: Yes, the size of the fingerprint is determined by the resolution of the 3D printer. It is technically difficult to further reduce the size.

Q5. The frequency resolution is determined to be 0.1 Hz. The true frequency-resolution should be even higher because the authors tried only frequency difference of 0.1 Hz. Sometimes people use the FWHM of the peaks as the resolution.

Response: We thanks the reviewer for the suggestion. We have updated the frequency-resolution in the revised manuscript, and the frequency-resolution is determined to be ~0.02 Hz based on the FWHM of the peaks.

Modification: Line 18-21, Page 9

“.....The frequency-resolution is determined to be ~0.02 Hz (or ~0.005% at 400 Hz), identified by the full width at half maximum (FWHM) of the peaks. Such a high frequency-resolution allows the slip-sensor to identify surface textures with close feature spacings.”

Q6. There is a problem in the format of references (Ref. 19 and 20) in page 5. Please correct it.

Response: Done!

Q7. In figure 1a, I suggest that the authors change the expression “rough or fine” to “rough or smooth”.

Response: Thanks for the suggestion! We have changed the expression “rough or fine”

to “rough or smooth” in Fig. 1a in the revised manuscript.

Reviewer #2 (Remarks to the Author)

The authors propose an artificial sensory system that can respond to both static and dynamic stimuli with high spatial resolution and high frequency resolution. By leveraging machine learning analytics, high-precision discrimination of textile surface textures has been demonstrated for robotic applications. Although the sensor in this paper shows a great advantage in sensitivity, response relaxation time, and corresponding frequency range compared to other capacitive sensors, the novelty of its system level application is not particularly obvious compared to current state-of-art works, which needs to be strengthened. Thus, I suggest major revision with the following comments properly considered:

Response: We acknowledge the reviewers’ comments on the system level of this work. The article has been thoroughly revised based on the reviewers’ suggestions. We believe that such revisions have significantly improved this work.

Q1. For the sensor structure, the periodic domes are designed with a diameter of 200 μm and a height of 55 μm . The choice of this parameter or the related optimization data/discussion are suggested to be mentioned in the text.

Response: We thank the reviewer for the suggestion. The specific dimensions of the periodic dome, including its diameter and height, are determined by a trade-off between printing resolution of the 3D printer and the thickness of the device. Specifically, the thickness of the sensor should be made to be as thin as possible to achieve a high flexibility, whereas a limited thickness makes the introduction of graded microstructures difficult. We finally select a diameter of 200 microns and a height of 55 microns. We have emphasized this point in the revised manuscript.

Modification: Line 2-4, Page 6

“.....The specific dimensions of the periodic domes or finer protrusions, including their diameters and heights, are determined by a trade-off between the fabrication resolution and the thickness of the device.”

Q2. Typos and errors, e.g., “diamater” in “...with a diamater of 200 μm and a height of 55 μm ,”. The authors need to check through the main text again carefully.

Response: The word “diamater” has been corrected to “diameter”, and we have thoroughly checked the typos and grammar problems in the revised manuscript.

Q3. For the spatiotemporal resolution test of the slip-sensor, the authors fixed the sliding rate to 1.0 mm s^{-1} . However, the contact force/pressure may also affect signal

amplitude during sliding, which needs to be mentioned or discussed in the text.

Response: We thank the reviewer for the suggestion. Indeed, the contact pressure will affect signal amplitudes. As shown in the **Fig. R1** (or revised **Supplementary Fig. 7**), the signal magnitude increases with contract pressure because of the stronger interaction between the slip-sensor and the microstructure.

Fig. R1. Signals to the microstructures with a spacing of $15 \mu\text{m}$ at a sliding rate of $1.0 \text{ mm}\cdot\text{s}^{-1}$ under contact pressures of 2, 10, and 20 kPa.

Modification: From Line 22, Page 8 to Line 2, Page 9

“.....In addition, signal magnitude increases with contact pressure due to the stronger interaction between the slip-sensor and the microstructure (Supplementary Fig. 7).”

Q4. In Fig. 4, the authors demonstrated that the developed artificial sensory system can achieve high-accuracy textile recognition with a given sliding rate. However, in practical applications, the sliding rate is not always well maintained. What is the performance of this perception system at a relatively random sliding rate, e.g., to simulate the relatively random touch done by a human hand?

Response: A great point! In the revised manuscript, we have added texture recognition at random sliding rates. We show that system can achieve an average accuracy of 98.6% at random sliding rates (updated **Fig. 4h** and **Fig. 4i**).

Modification: From Line 18 Page 12 to Line 4, Page 13

“The human finger touch objects at a random sliding rate, we thus conducted texture recognition at a random sliding rate. We fixated the slip-sensor on an index finger of a human subject and unconsciously slid the sensor over the textiles with unknown contact pressure and sliding rate. Here, we collected signals for the whole interaction between the sensor and the textile: before contacting, finger touching, finger sliding over the textile, and finger withdrawing (Fig. 4h and Supplementary Fig. 15).

We collected a dataset comprised of 4200 instances for the 20 textiles, with 210 entries per category. The original dataset was divided into training and test sets at a ratio of 6:4, and we used the Random Forest algorithm for classification. An average recognition accuracy of 98.6% was achieved (Fig. 4i), revealing the high robustness and reliability of our sensory system in texture recognition.”

Q5. In Fig. 2, the authors demonstrated that the proposed sensor achieved the static and pressure detection simultaneously, and mimic the SA and FA of biological sensory system. However, for the textile recognition shown in Fig. 4 and Fig. 5, the authors only use the dynamic signal as the input of perception model. Actually, some works have proved that the fusion of static and dynamic features can obtain better texture recognition performance, i.e., Nano letters 19.5 (2019): 3305-3312. Such data is suggested to be added to make the story of the whole article seem more complete.

Response: Thanks for the valuable suggestion. We have cited this work and conducted texture recognition that considers both static and dynamic features, and the result verifies a higher recognition accuracy (Fig. 4i).

Q6. Many recent works have shown the possibility of simulating SA and FA for texture recognition under flexible sensory platform for HMI and robotic applications. A comparison table is needed to compare this work with other state-of-art works to further highlight the novelty of this paper.

Response: We thank the reviewer for the constructive suggestion. In the revised manuscript, we have added **Supplementary Table 2** to compare our sensory system with existing sensory systems that simulate SA and FA receptors for texture recognition.

Modification: Line 4-9, Page 14

“Existing work has reported several sensory systems for texture or material recognition using two types of sensors (e.g., piezoresistive and piezoelectric sensors, or piezoresistive and triboelectric sensors) to simulate SA and FA receptors^{25,26,46-49}. These sensory systems, however, require two sensors integrated together with two sets of data acquisition systems (Supplementary Table 2). In contrast to these sensory systems, our sensory system can achieve a high recognition accuracy using a single sensor, while the system is simplified and robust.”

Q7. Some recent interesting works can be referred in introduction to broaden the view of readers, e.g., Science Advances 8.31 (2022): eabq2521; ACS Nano, vol. 17, no.5, 4985-4998, 2023; ACS Nano, vol. 17, no.2, 1355-1371, 2023; Adv. Energy Mat., vol. 13, no.1, 2203040, 2023; Nat. Commun., vol. 13, 5224, 2022.; Applied Physics Rev, vol. 7, no. 3, 031305, 2020.

Response: We have cited these works in the *Introduction* of the revised manuscript to

better serve the community.

Reviewer #3 (Remarks to the Author):

The authors present a novel sensor based on tunable electric double layers (EDLs) that have a nanoscale charge separation which translates mechanical stimuli into capacitive signals. The sensor responds to static and dynamic stimuli and shows impressive performance *i.e.* sensitivity up to 519 kPa^{-1} , spatial resolution down to $\sim 15 \text{ }\mu\text{m}$ in spacing and $6 \text{ }\mu\text{m}$ in height, frequency vibrations measurement sensitivity up to 400 Hz, frequency-resolution of 0.1 Hz, total response-relaxation time of $\sim 2.4 \text{ ms}$. The fingerprint has a bio-mimetic structure and geometry. The authors present results on the recognition of 20 different commercial textiles with high accuracy and a portable system with a robotic hand which integrates the sensor, a PCB for interfacing with a PC and for real-time graphic interface for surface classification.

Strong points:

Novel iontronic sensor which measures both static and dynamic pressure/mechanical stimuli with low (for a capacitive sensor) response-relaxation time, high frequency bandwidth and frequency resolution.

Response: We greatly appreciate the reviewer for the evaluation on the high performance of our sensor.

Weak points:

Q1. Missing relevant information on the experimental setup (see other comments).

Response: We thank the reviewer for pointing out the lack of details in some of experimental setups. Per the reviewer's suggestions, we have supplemented the method for the computation of sensitivity, the design of circuit board, and other detailed information of the sensory system.

Q2. The machine learning approach looks overestimated: Fig. 4.e shows that, with a proper transformation, different clusters are almost separated, a simpler algorithm (e.g. SVM) could perform the task. The authors should compare the classification results with a reference algorithm in the state of the art.

Response: We thank the reviewer for the comments on the machine learning method. In fact, it is crucial to consider uncertainties in data acquisition, such as variations in contact pressure, tilt angle of the textile, and the movement of yarns in the textiles during repeated rubbing. These factors affect the extraction of features from the raw signals. Therefore, we need a model that can handle such complex situations.

Here, our model can not only achieve a high classification accuracy in dealing with

existing datasets, but also be promising to deal with newly collected and real-time data, considering the aforementioned uncertainties. That is, our method exhibits high robustness and adaptability.

Q3. The comparison with the human capabilities in textile recognition by touch exploration is not meaningful, I expect that an artificial system overcomes human performance in many different ways. In the proposed results, the artificial system applies the stimulus in a very controlled way which oversimplifies the task. This part of the paper is not meaningful and should be removed.

Response: We thank the review for the suggestion. The part of texture recognition by human subjects is removed from the revised manuscript. Instead, we have added a new part on the recognition with random sliding rates.

Q4. Presentation in some points is obscure, more details and a clear presentation must be supplied in many parts. In the last part, the paper presents a touch-based object recognition experiment. This part should be revised as the task performed is the recognition of surface objects by sliding the sensor on the surface itself.

Response: We thank the reviewer for pointing out the problem in presentation. In the revised version, the part for the recognition of surface objects has been removed from the manuscript since this part is not clear and not solely related to textures. Please also see our *Response* to Q29.

Q5. The read-out electronic circuit of the portable system must be clearly presented.

Response: We thank the reviewer for the constructive suggestion. In the revised manuscript, we provide the details on the readout circuit board and the portable sensory system.

Modification: On Line 12-16, Page 11, and Line 2-13, Page 18, we add a detailed description of the readout circuit.

On Line 12-16, Page 11: “The circuit board consists of five parts: a power supply module, a microcontroller module (STM32) serving as the central processing unit to process data and make decision, an input/output interface module for the communication with external devices, a signal processing module responsible for conditioning internal signals, and a 24-bit analog-to-digital conversion (ADC) module for sampling signals (Supplementary Fig. 12).

On Line 2-13, Page 18: “The design principle of the circuit is as follows. First, a 12-bit digital-to-analog converter inside the STM32 microcontroller is programmed to generate a stable sine wave signal as the excitation source for the measurement. This excitation signal is applied to the capacitor to be measured, causing a current to flow

through the capacitor. During this process, the capacitor exhibits a fixed resistive characteristic known as capacitive reactance, which is proportional to the capacitance value. Next, a reactance-to-voltage conversion circuit is used to transform the capacitive reactance into a voltage signal that is proportional to the capacitance value. Subsequently, the output voltage signal is processed through a low-pass filter to obtain a DC voltage. Finally, this DC voltage is sampled using a 24-bit ADC, and the STM32 microcontroller calculates the value of the capacitance being measured (Supplementary Fig.12).

The circuit board had dimensions of 5.5 cm in length and 3.5 cm in width (inert in Supplementary Fig.12), which are determined based on the functional requirements and component quantity.”

From Line 11, Page 13 to Line 3, Page 14, we add the details on the portable sensory system: “.....a circuit board for collecting sensing information of textures and sending real-time data to a PC via USB wired transmission, the aforementioned machine learning method for classifying the textures, and a user interface for visualizing the output results (Fig. 5a).

In the real-time sensory system, all 2000 data sets collected were used for training, while the test set is independent data collected by the circuit board in real time. By analyzing the real-time data, features were extracted and classification models were applied to make immediate predictions or recognitions. Real-time recognition enables rapid decision making and faster feedback based on streaming data, making it suitable for time-sensitive applications or scenarios that require immediate response. Our experimental results showed that the inference time was below 20 ms, validating the real-time feasibility of our sensing system. With the machine learn-based classifier, we can identify the textiles in the signals collected in real time and display the confidence of the recognition as well as the microscopic morphology of the textiles identified on a real-time visual user interface. Figure 5b shows the implementation of two real-time sensing systems, showing the high confidence when a prosthetic hand with an integrated slip-sensor touches textiles. Of particular note is that the real-time system showed an average accuracy of 98.5% for these 20 textiles (Supplementary Movie 1).”

Q6. The claim in the Introduction “the selection of low-viscosity ionic material together with the microstructural design allow the sensor to rapidly respond to high-frequency vibrations” is not sufficiently motivated/demonstrated in the paper. I see two different points here: 1) the fingerprint arrangement which is biomimetic and improves the spatial resolution; 2) the ionic-material as insulating layer of the sensing capacitor which allows to achieve high frequency. The authors should better explain these two points.

Response: In the *Introduction*, the sentence “the selection of low-viscosity ionic material together with the microstructural design allow the sensor to rapidly respond to high-frequency vibrations” points out the impact of viscosity and microstructures of

ionic materials on response time.

For the first point, the fingerprint structure plays a crucial role in texture recognition. We used a PDMS fingerprint to simulate human fingerprints and to capture vibrational stimuli during the interaction between the sensor and textiles. The size of the fingerprint tip determines the spatial resolution of the sensor. A smaller size of fingerprint tip down to microns can fill in smaller gaps of surface textures to effectively interact with the textures.

For the second point, we utilized a low viscosity ionic gel to minimize adhesion strength at the electrode-ionic gel interface, resulting in a substantial increase of response-relaxation time. In addition, we introduced a graded microstructure consisting of microdomes with finer protrusions on the ionic gel, which reduces the contact area between the electrode and the ionic gel. A smaller contact area induces a further reduction in the interfacial adhesion energy (see **Fig. 2j**), and thus leading to shortened response-relaxation time. In addition, the densely distributed graded microstructure can behave like springs to enable rapid elastic recovery and release of strain energy, reducing the response-relaxation time.

Besides, the smaller anions (H^+) and inorganic anions in the PVA- H_3PO_4 gels can also contribute to fast ion migration and thus a rapider response-relaxation speed compared with traditional ionic liquid. We have added the explanation in the revised manuscript.

Modification: Line 7-9, Page 8

“.....In addition, the ionic radii of hydrogen ions and inorganic anions in the PVA- H_3PO_4 gel enable faster ion migration, contributing to a rapid response-relaxation speed as well.”

Q7. Which are the advantages of using ionic-material with respect to other materials?

Response: The ionic material forms an electric double layer (EDL, for which the charge separation is on nanoscale) with an electronic material (electrode). The capacitance of an EDL-based capacitor is a few orders of magnitude higher than that of conventional capacitors. This leads to ultrahigh sensitivity and high signal magnitude.

Q8. The advantages of the “microstructured surface with two-levels of structures” (see page 5) should be better discussed and demonstrated. The measurement of the static capacitance versus normal mechanical stimulus should be reported. The way the microstructured surface affects the capacitance value upon loading should be introduced.

Response: The advantages of the microstructured surface have been discussed in the

revised manuscript. We have also added a supplementary figure (see **Fig. R2** Or **Supplementary Fig. 4**) to visualize the contact process that two levels of microstructures are involved.

Fig. R2. The Capacitance of the slip-sensor at different pressures. The inset pictures show the interfacial contact area under different pressures.

Modification: Line 11-18, Page 6

“The high sensitivity is attributed to the subtle change in microstructured EDL interface upon loading. Before applying a pressure, the presence of air gap prevents the contact between the electrode and the ionic gel, resulting in a low initial capacitance (C_0) of ~ 8 pF. When a pressure is applied, the smaller protrusions of the ionic gel begin to contact with the electrode, and the signal increases sharply because of the increasing EDL capacitance. As the pressure further increases, the larger microdomes are involved in the contact and the capacitance remains increasing (Supplementary Fig. 4). Therefore, such two-level microstructures increase the sensitivity and extend the working range of the sensor.”

Q9. The authors must explain the reason why “the hierarchical microstructures of the ionic layer help improve the sensitivity and reduce the response-relaxation time of sensors.” Page 6, top.

Response: Thanks for the comments regarding the effect of the hierarchical microstructure on sensitivity and response-relaxation speed. The reason that the microstructures can improve sensitivity has been explained in our *Response* to Q8. We have added text in the revised manuscript to explain the effect of hierarchical microstructure on response-relaxation speed (Line 4-7, Page 8).

Introducing microstructure can enhance the response-relaxation speed of sensors due to that the microstructures can quickly store and release energy for elastic recovery (*Nat. Mater.* 2010 **9**, 859-864). The smaller contact area caused by the microstructures further reduces the adhesion energy of the interface (*Nat. Commun.* 2019, **10**, 4405),

and increases the energy release rate of the interface, thereby leading to an improved response-relaxation speed.

Q10. The section structure with the thickness of the sensor various layers must be reported.

Response: The thickness of each layer has been provided in the structure diagram of the slip-sensor, as illustrated in revised **Fig. 2a**.

Q11. Due to the total response-relaxation time of ~2.4 ms which is very impressive for a capacitive sensor, the corresponding maximum frequency that can be measured by the sensor is 416 Hz which overcomes capacitive sensors at the state of the art. The authors should explain how the measurement setup of Fig. 2.d, if the measurements were static or dynamic, they should report the values of ΔC and C_0 , it is not clear to me how they computed the sensitivity.

Response: We appreciate the reviewer for the recognition of the response-relaxation speed and frequency bandwidth. We have added the details on the measurement in the *Characterization and Measurements* of the revised manuscript.

Modification: From Line 16-23, Page 16

“.....The sensitivity was defined as $S = \delta (\Delta C/C_0) / \delta P$, where P represents the applied pressure, and ΔC is the difference between the measured capacitance C and the initial capacitance C_0 (~8 pF). Increased dynamic pressures up to 100 kPa were applied to the sensor, and the corresponding peak capacitance value for each pressure was recorded using the LCR meter. The slope in the $\Delta C/C_0$ - P curve represented the sensitivity value of the sensor.

The response-relaxation time was tested by rapidly applying and withdrawing a pressure of ~50 kPa to the sensor using manual pulling on a flat metal post with the diameter of 3 mm. The time for the rising edge represented the response time, and the time for the descending edge was the relaxation time.”

Q12. Fig. S2 should report the input stimulus versus time; 1000 cycles are not a relevant value to evaluate signal drift. I would increment the number of cycles by one order of magnitude at least.

Response: According to the reviewer’s suggestion, the number of compression cycles was extended to 10,000, with a total test time of ~14.5 h (**Fig. R3** or **Supplementary Fig. 2**).

Fig. R3. Change in capacitance over 10,000 loading-release cycles under a peak pressure of 100 kPa, with a total test time of ~14.5 h.

Q13. The authors should make experiments to evaluate the hysteretic behavior of the sensor.

Response: The hysteretic behavior of the sensor has been added in the revised manuscript, and negligible hysteresis was observed, as shown in **Fig. R4** or revised **Supplementary Fig. 3**.

Fig. R4. Hysteresis curve of the slip-sensor during a loading-unloading cycle.

Modification: Line 9-10, Page 6

“.....Furthermore, the sensor exhibits low hysteresis by loading a maximum pressure of 100 kPa and releasing (Supplementary Fig. 3).”

Q14. Which is the sensor output sampling rate?

Response: The signals of the sensor were recorded using either a high-speed LCR digital bridge (TH2840B, Tonghui) or a digital circuit board. The sampling frequency is 1600 Hz using an LCR meter for frequency bandwidth measurement. The circuit

board uses a different sampling frequency of 1000 Hz.

The information has been added in the *Methods* part (Line 12-15, Page 16).

Q15. What are piezocapacitive sensors (page 6, top)?

Response: “Piezocapacitive sensors” refers to capacitive sensors for which capacitance changes with pressure. This term has already been used in many published papers (*e.g.*, *Nat. Commun.* **2020** 11, 5747; *Sci. Rep.* **2020**, 10, 12666; *Nat. Commun.* **2020**, 11, 209; *Nanoscale* **2021**, 13, 6076-6086; *Nano-Micro Lett.* **2022**, 14, 141; *Nat. Commun.* **2022**, 13, 5839).

Q16. The thickness of the ionic gel is 120 microns, the ionic gel implements the insulating layer of the capacitor, which is the dielectric constant of the material? The capacitance value is in the order of nF (see Fig. 2.e), which is the capacitance value per unit area? When the mechanical stimulus is applied, the thickness of the ionic gel decreases, and the capacitance increases, which is the maximum capacitance value which has been measured during experiments? Which is the corresponding ionic gel thickness?

Response: The iontronic sensors are based on the electric double layer, for which the charge separation is ~1 nm. The capacitance of iontronic sensors is determined by the contact area of the iontronic interface instead of the thickness of the ionic gel. As a result, the dielectric constant of the ionic gel, a frequency-sensitive parameter, is often not measured.

The capacitance density can reach ~330 nF·cm⁻² at 100 kPa. The maximum capacitance is 126 nF, which is measured at 100 kPa. Since in iontronic sensors the capacitance is not dependent on thickness, we did not measure the thickness of the ionic gel.

Q17. In Fig. 2.f, the amplitude value must be reported on the y-axis. Which is the input stimulus amplitude? Which is the measurement setup?

Response: Thanks. We have given the amplitude value of the y-axis. In addition, the details of the measurement setup are described in the *Methods* part (On Line 2-3, Page 17).

Q18. The sentence at page 7 “The interfacial adhesion behavior between the electrode and the ionic gel determines the relaxation time of the sensor” should be better explained also with experimental evidence.

Response: This is because iontronic sensing is an interfacial behavior—the capacitance

value is in direct proportion to the interfacial contact area. Therefore, the interfacial adhesion determines the relaxation time.

We have changed the sentence to “The interfacial adhesion behavior between the electrode and the ionic gel determines the relaxation time of the sensor because the contact area is proportional to the capacitance value.” On Line 19-20, Page 7.

Q19. The motivations of the frequency resolution of the sensor of 0.1 Hz should be clearly discussed.

Response: The high frequency resolution allows the sensor to identify surface textures with close feature spacings. A sensor exhibits a wide frequency detection range (bandwidth) may not discriminate close frequencies in the detection range. Therefore, the determination of frequency resolution is important.

Modification: Line 18-21, Page 9

“.....The frequency-resolution is determined to be ~0.02 Hz (or ~0.005% at 400 Hz), identified by the full width at half maximum (FWHM) of the peaks. Such a high frequency-resolution allows the slip-sensor to identify surface textures with close feature spacings.”

Q20. The electronic interface circuit for capacitive variation measurement must be presented. Fig. S9 does not help in understand the read-out circuit.

Response: Thanks for the suggestion. We have provided the details of the readout circuit board.

Modification: On Line 12-16, Page 11, and Line 2-13, Page 18, we supplement a detailed description of the readout circuit board.

On Line 12-16, Page 11: “The circuit board consists of five parts: a power supply module, a microcontroller module (STM32) serving as the central processing unit to process data and make decision, an input/output interface module for communication with external devices, a signal processing module responsible for conditioning internal signals, and a 24-bit analog-to-digital conversion (ADC) module for sampling signals (Supplementary Fig. 12).”

On Line 2-13, Page 18: “The design principle of the circuit is as follows. First, a 12-bit digital-to-analog converter inside the STM32 microcontroller was programmed to generate a stable sine wave signal as the excitation source for the measurement. This excitation signal was applied to the capacitor to be measured, causing a current to flow through the capacitor. During this process, the capacitor exhibited a fixed resistive characteristic known as capacitive reactance, which was proportional to the capacitance value. Next, a reactance-to-voltage conversion circuit was used to transform the capacitive reactance into a voltage signal that is proportional to the capacitance value. Subsequently, the output voltage signal was processed through a low-pass filter to

obtain a DC voltage. Finally, this DC voltage was sampled using a 24-bit ADC, and the STM32 microcontroller calculated the capacitance value.

The circuit board had dimensions of 5.5 cm in length and 3.5 cm in width (inset in Supplementary Fig.12), which were determined based on the functional requirements and component quantity.”

Q21. Miniaturization of the read-out PCB should be discussed.

Response: The circuit board has dimensions of 5.5 cm in length and 3.5 cm in width. The dimensions may be further reduced by optimizing the components and wiring layouts.

Modification: Line 11-13, Page 18

“The circuit board had dimensions of 5.5 cm in length and 3.5 cm in width (inset in Supplementary Fig.12), which were determined based on the functional requirements and component quantity.”

Q22. The organization of the data set e.g. time duration of each sample, sliding speed of each sample, static pressure of each sample, must be provided by the authors.

Response: The specific parameters such as sliding rate, sliding distance, and static pressure are provided in *Methods* of the revised manuscript.

Modification: Line 16-18, Page 17

“The data for the 20 textiles were collected at a sliding distance of 40 mm and a sliding rate of $2 \text{ mm}\cdot\text{s}^{-1}$, with a total of 2000 data sets. A static pressure of $\sim 50 \text{ kPa}$ was applied to the sensor before sliding. The texture cognition test at random sliding rates was conducted by fixing the slip sensor on the subject's index finger and unconsciously sliding the sensor onto the textile with an estimated sliding rate of $0\text{-}30 \text{ mm}\cdot\text{s}^{-1}$. The details about the recognition of microstructure by human subjects was provided in the *Supplementary Information*.”

Q23. In Fig. 4.g are samples of different sliding velocities collected? Or only one sliding velocity has been used? It looks that experiments have been conducted with a constant sliding velocity. What happens with a dataset collecting samples with different sliding velocities?

Response: A great point! **Fig. 4g** and Supplementary Fig. 14 show the recognition accuracies at constant sliding rates of 2 and $40 \text{ mm}\cdot\text{s}^{-1}$, respectively. In the revised manuscript, we have added a texture recognition test at random sliding rates, which shows an average recognition accuracy of 98.6% (updated **Fig. 4h** and **Fig. 4i**).

Modification: From Line 18, Page 12 to Line 4, Page 13

“The human finger touch objects at a random sliding rate, we thus conducted texture

recognition at a random sliding rate. We fixated the slip-sensor on an index finger of a human subject and unconsciously slid the sensor over the textiles with unknown contact pressure and sliding rate. Here, we collected signals for the whole interaction between the sensor and the textile: before contacting, finger touching, finger sliding over the textile, and finger withdrawing (Fig. 4h and Supplementary Fig. 15).

We collected a dataset comprised of 4200 instances for the 20 textiles, with 210 entries per category. The original dataset was divided into training and test sets at a ratio of 6:4, and we used the Random Forest algorithm for classification. An average recognition accuracy of 98.6% was achieved, revealing the high robustness and reliability of our sensory system in texture recognition (Fig. 4i).”

Q24. Sentence in the Introduction: “However, artificial sensors often lack the ability or perform insufficiently to perceive, recognize, and explore the real world upon touching the target objects.”, exploration is an active action which is not implemented by sensors, I suggest removing it.

Response: Done as suggested.

Q25. The information about experimental setup and dataset structure is not sufficient; the authors should supply all the features of samples used to build the dataset.

Response: We have added the information in the revised.

Modification: On Line 3-13, Page 12, we provide detailed information about machine learning. From Line 11, Page 13 to Line 3, Page 14, we add the experimental set for portable sensing system

On Line 3-13, Page 12: “We use a Bagging ensemble learning approach to solve the classification problem, which improves the generalization capability of the model and the overall classification performance. The classification models (or classifiers) used in ensemble include K-nearest neighbors, random forests, logistic regression algorithms, and decision trees. In order to accurately distinguish between different textures, a variety of signal features are extracted, such as statistical, frequency domain, autoregressive, wavelet transform, and time domain features. There are 20 categories and dozens of features in our dataset, and Fig. 4f shows four categories and two features for simplified illustration. Each category has 100 sets of data, which were divided into five blocks. We selected one block at a time as the testing set and the rest as the training data, and iterated the prediction results in the testing set for several times. By combining the predictions of each base classifier through voting, more accurate and robust overall classification results can be achieved.”

From Line 11, Page 13 to Line 3, Page 14: “..... a circuit board for collecting sensing information of textures and sending real-time data to a PC via USB wired

transmission, the aforementioned machine learning method for classifying the textures, and a user interface for visualizing the output results (Fig. 5a).

In the real-time sensory system, all 2000 data sets collected were used for training, while the test set is independent data collected by the circuit board in real time. By analyzing the real-time data, features were extracted and classification models were applied to make immediate predictions or recognitions. Real-time recognition enables rapid decision making and faster feedback based on streaming data, making it suitable for time-sensitive applications or scenarios that require immediate response. Our experimental results showed that the inference time was below 20 ms, validating the real-time feasibility of our sensing system. With the machine learn-based classifier, we can identify the textiles in the signals collected in real time and display the confidence of the recognition as well as the microscopic morphology of the textiles identified on a real-time visual user interface. Figure 5b shows the implementation of two real-time sensing systems, showing the high confidence when a prosthetic hand with an integrated slip-sensor touches textiles. Of particular note is that the real-time system showed an average accuracy of 98.5% for these 20 textiles (Supplementary Movie 1).”

Q26. The authors should evaluate results with a dataset of samples collected at different velocities and different static pressure. I believe that if the sliding velocities and static pressure are constant the operation of the Machine Learning algorithm is over simplified as shown in 4.e where clusters of different textiles are clearly identified and do not overlap. This is in my opinion the very high classification accuracy that is achieved. Humans cannot precisely control the sliding velocity as such their performance are not that high if compared with those of a machine learning algorithm with data collected with precisely controlled operating conditions.

Response: We thank the reviewer for the suggestion. In the revised manuscript, we have added texture recognition under random sliding rate and pressures. The details are provided on **Line 18-23, Page 12** of the revised manuscript. Please also see our *Response to Q4 and Q23*.

Q27. The inference time, i.e. the time needed for output the classification result should be reported in the paper.

Response: The inference time is below 20 ms, which has been provided in the revised manuscript.

Modification: Line 20-21, Page 13

“Our experimental results showed that the inference time was below 20 ms, validating the real-time feasibility of our sensing system.”

Q28. The training and test sets, the related operating conditions (see Fig. 5), should be clearly reported in the paper. Which are the differences with respect to what reported in

Fig. 4? Which is the meaning of the following sentence “all the 2000 sets of data were used as the training set, and we extract features for each textile to further establish a classification model”, which features are extracted? Which classification model has been used?

Response: We have elaborately illustrated the training set, test set, the related operating conditions for the portable sensory system (Fig. 5) in the revised manuscript.

Non-real-time texture recognition (Fig. 4) and real-time texture recognition in the portable sensory system (Fig. 5) both use a Bagging ensemble learning approach to solve the classification problem, which improves the generalization capability of the model and the overall classification performance. The classification models (or classifiers) used in ensemble include K-nearest neighbors, random forests, logistic regression algorithms, and decision trees. In order to accurately distinguish between different textures, a variety of signal features are extracted, such as statistical, frequency domain, autoregressive, wavelet transform, and time domain features.

We collected 100 sets of data for each of the 20 textiles with a total of 2000 data sets generated by the sensor. For model development, we used 80% of these 2000 data sets as the training set and reserved the remaining 20% as the test set, for which the recognition results were presented through a confusion matrix (Fig. 4). This approach represents non-real-time recognition, where the classification or recognition of data occurs after the data collection process is complete. It typically involves offline processing and analysis, allowing for more computational resources and time to achieve accurate recognition results. Therefore, we further established a real-time sensing system based on these 2000 data sets.

The difference between the real-time sensory system (Fig. 5) and the aforementioned approach lies in the composition of the training set and the test set. In the real-time sensory system, all 2000 data sets collected were used for training, while the test set is independent data collected by the circuit board in real time. By analyzing the real-time data, features were extracted and classification models were applied to make immediate predictions or recognitions. Real-time recognition enables rapid decision making and faster feedback based on streaming data, making it suitable for time-sensitive applications or scenarios that require immediate response.

Modification: On Line 3-8, Page 12: “We use a Bagging ensemble learning approach to solve the classification problem, which improves the generalization capability of the model and the overall classification performance. The classification models (or classifiers) used in ensemble include K-nearest neighbors, random forests, logistic regression algorithms, and decision trees. In order to accurately distinguish between different textures, a variety of signal features are extracted, such as statistical, frequency domain, autoregressive, wavelet transform, and time domain features.”

On Line 15-20, Page 13: “In the real-time sensory system, all 2000 data sets collected were used for training, while the test set is independent data collected by the

circuit board in real time. By analyzing the real-time data, features were extracted and classification models were applied to make immediate predictions or recognitions. Real-time recognition enables rapid decision making and faster feedback based on streaming data, making it suitable for time-sensitive applications or scenarios that require immediate response.”

Q29. It is claimed that the system can identify macroscale objects (page 13), my understanding is that the sensor can classify different surfaces by sliding the sensorized finger on the object surface. The operating conditions of such experiment must be clearly reported. Discussion on results is confuse, the assessment is not convincing.

Response: Thanks for pointing out the problems. We have removed this part from the manuscript since it is not a necessary part of the work and may cause confusion.

Q30. The authors must compare their results with the ones of: Zhang, J., Yao, H., Mo, J. et al. Finger-inspired rigid-soft hybrid tactile sensor with superior sensitivity at high frequency. Nat Commun 13, 5076 (2022). <https://doi.org/10.1038/s41467-022-32827-7>

Response: We have compared our slip-sensor sensor with the one mentioned by the reviewer.

The revised *Ref. 33* is the one mentioned by the reviewer.

Modification: Line 1-4, Page 7

“Such a rapid response-relaxation process enables the sensor to effectively respond to high-frequency vibrations up to 400 Hz, as shown in the time-dependent capacitance signals and the corresponding Fourier transform spectra (Fig. 2f). The response-relaxation time is almost two orders of magnitude shorter than that of existing capacitive sensors, and comparable to that of a rigid-soft hybrid sensor³³.

REVIEWER COMMENTS

Reviewer #1 (Remarks to the Author):

The authors made sufficient revisions and the paper is recommended for publication now.

Reviewer #2 (Remarks to the Author):

In the revised manuscript, the authors have addressed the majority of my comments, leading to an improvement in the quality of this manuscript. However, there remain certain details that require modification to enhance the scientific rigor and persuasiveness of the entire paper. I think this manuscript can be accepted after a minor revision, accompanied by the following comments:

Related to my previous comments:

Previous Q1: Although the authors have provided reasons for selecting 200 μm and 55 μm based on the difficulty of preparation and the flexibility of the device, the dome structure is crucial in detecting the vibrations generated by the surface fingerprint pattern. At the same time, the authors have chosen a spacing of 350 μm and a height of 260 μm for the fingerprint pattern. Hence, it raises the question of which of these parameters is more significant in affecting surface texture detection? For instance, it is later mentioned that the sensor can recognize a structure with a minimum period of 231 μm . Is this because this value exceeds the 200 μm diameter of the dome? If so, what is the importance of the surface fingerprint pattern?

Previous Q2: No further questions.

Previous Q3: If the amplitude changes in lockstep with force during the sliding process, can a high recognition accuracy still be achieved if the applied force is not controlled at a consistent and stable value?

Previous Q4: In the newly added data, recognition accuracy with random speed and force on a human hand appears even higher than under precise control. The authors should provide additional explanation to elucidate this experimental result. Furthermore, attributing randomness solely to being “unconsciously” caused lacks sufficient scientific support. Also, the sliding habits of the same individual across multiple data sets would likely remain consistent. It is suggested that the authors might utilize instruments like IMU to quantify this randomness through sensing data.

Previous Q5-Q7: No further questions.

New comment to be addressed:

New Q1: The scientific connection between the frequencies tested in Figure 2 and the

vibrations used for texture recognition later in the text appears insufficiently tight. This is because the frequencies tested in Figure 2 are generated by vertical pressure, while the frequencies used for identification are produced by horizontal frictional forces. These may be more appropriately linked through common characteristics, such as the device's response time, rather than directly associating them through frequency.

Reviewer #3 (Remarks to the Author):

Page3, Line 18, artificial not artificials

Page 5, lines 7-8, sensor signal not signal sensor

Supplementary Fig. 1 is a replica of Fig. 2.b,c, I suggest to remove it

Fig. 3.e, which is the applied stimulus? Which is the measurement setup?

I suggest replacing Video 1 and Fig. 5 with experimental results at variable sliding velocity.

The Discussion section looks as a conclusion section, the authors should revise the text accordingly to the section title.

Page 13, lines 16-17. Which are the extracted signal features? The authors should introduce a Table with all the extracted and used features.

The authors should evaluate and report in the paper, the signal-to-noise ratio and the effective number of bits of the sensor output signal.

Response to reviewers for manuscript NCOMMS-23-15409B

Reviewer #2 (Remarks to the Author):

In the revised manuscript, the authors have addressed the majority of my comments, leading to an improvement in the quality of this manuscript. However, there remain certain details that require modification to enhance the scientific rigor and persuasiveness of the entire paper. I think this manuscript can be *accepted after a minor revision*, accompanied by the following comments.

Response: The authors appreciate the reviewer for the positive evaluation to our revised version.

Related to my previous comments:

Q1: Although the authors have provided reasons for selecting 200 μm and 55 μm based on the difficulty of preparation and the flexibility of the device, the dome structure is crucial in detecting the vibrations generated by the surface fingerprint pattern. At the same time, the authors have chosen a spacing of 350 μm and a height of 260 μm for the fingerprint pattern. Hence, it raises the question of which of these parameters is more significant in affecting surface texture detection? For instance, it is later mentioned that the sensor can recognize a structure with a minimum period of 231 μm . Is this because this value exceeds the 200 μm diameter of the dome? If so, what is the importance of the surface fingerprint pattern?

Response: We use a spacing of 350 μm and a height of 260 μm for the fingerprint pattern to mimic human fingerprints—which often have a spacing of 300~500 μm and a height of 100~300 μm . We will further investigate the effect of fingerprint parameters on the classification in our further studies.

In fact, our sensor can detect structures with periods much smaller than 231 μm , as shown in Fig. 2b. The spatial resolution (the smallest period that our sensor can detect) is down to ~15 μm .

Q3: If the amplitude changes in lockstep with force during the sliding process, can a high recognition accuracy still be achieved if the applied force is not controlled at a consistent and stable value?

Response: Yes, we can still achieve a high recognition accuracy if the applied force is not stable. We show that when both pressure and sliding rate change, our system can still achieve a recognition accuracy of 98.9% (Fig. 4i).

Q4: In the newly added data, recognition accuracy with random speed and force on a human hand appears even higher than under precise control. The authors should provide additional explanation to elucidate this experimental result. Furthermore, attributing randomness solely to being “unconsciously” caused lacks sufficient scientific support. Also, the sliding habits of the same individual across multiple data sets would likely remain consistent. It is suggested that the authors might utilize instruments like IMU to quantify this randomness through sensing data.

Response: The authors thank the reviewer for pointing out the details. In the previous version, only five features were extracted for machine learning, owing to a certain level of consistency in the tactile data at a fixed sliding rate. For the case of random sliding rate and pressure, hundreds of features were extracted using a Python package (“Tsfresh” module), due to the diversity of the corresponding datasets, resulting in a slightly higher accuracy.

Here, in the latest version, we utilized “Tsfresh” module to extract hundreds of features for data with fixing sliding rate, achieving an exceptionally high accuracy of 100% (see the updated Fig. 4h). In the revised manuscript, the reasons for the high classification accuracy were outlined on **Lines 6-9, Page 12**.

We acknowledge the reviewer’s concern on randomness. Per the suggestion of the reviewer’s, we have used an inertial measurement unit (IMU) to quantify the randomness.

In our new dataset, each category involved random touches from three people in a 2:1:1 ratio, with a total of 400 sets data for per category, 40% of which were used for testing. We also recorded change in acceleration during sliding using an IMU. Because the sliding process occurred on a plane, we disregarded acceleration in the z-axis (the primary component in the direction of gravity). The chaotic acceleration in the x-y direction during multiple slides on each textile clearly illustrated the variability and unpredictability of the sliding rates (see the updated Supplementary Fig. 17-19).

Modification: On Lines 6-9, Page 12; Lines 18-24, Page 12

On Lines 6-9, Page 12, we have added “The output confusion matrix finally confirms an accuracy of 100.0% for textile recognition (Fig. 4g). The high recognition accuracy can be attributed to two aspects. First, the slip-sensor can sensitively capture the small differences between different textiles. Second, the tactile dataset for each textile is consistent at a fixed sliding rate. We extracted a multitude of features using

the “Tsfresh” module that helps achieve a high classification accuracy.”

On Lines 18-24, Page 12, we have added “Considering the potential consistency within the sliding habits of the same subject, we collected data of random sliding rates from three different subjects. Each category involved random touches from three individuals in a 2:1:1 ratio, resulting in a total of 400 sets data for each category and a total of 8000 sets for the 20 textile types, with 40% of the data reserved for testing. We used an inertial measurement unit (IMU) to record the acceleration during the sliding process. The evidently chaotic acceleration in the x - y plane confirms that the sliding rates of an individual continuously changes during sliding (Supplementary Fig. 16-18).”

Q5: The scientific connection between the frequencies tested in Figure 2 and the vibrations used for texture recognition later in the text appears insufficiently tight. This is because the frequencies tested in Figure 2 are generated by vertical pressure, while the frequencies used for identification are produced by horizontal frictional forces. These may be more appropriately linked through common characteristics, such as the device’s response time, rather than directly associating them through frequency.

Response: Many thanks for the suggestion. The response-relaxation time of our sensor is 2.4 ms, corresponding to a frequency bandwidth of 416 Hz ($1000/2.4$), and this agrees well with our experimental result (frequency bandwidth of 400 Hz).

During the horizontal sliding of the sensor, the artificial fingerprints interact with the surface textures. There will be vibrations along both the horizontal and the vertical directions during the interaction between the fingerprints and the object surface structures (**Fig. R1**). The two vibrations share the same frequency since the process is considered as linear.

Fig. R1 Schematic diagram of vibrations generated by the interaction of the slip-sensor with the surface structure of the object.

Reviewer #3 (Remarks to the Author):

Q1. Page3, Line 18, artificial not artificials; Page 5, lines 7-8, sensor signal not signal

sensor.

Response: Done as suggested.

Q2. Supplementary Fig. 1 is a replica of Fig. 2. b,c, I suggest to remove it.

Response: We thank the reviewer for the suggestion. Sorry to make these two figures confusing because we did not clearly describe their difference. In fact, Supplementary Fig. 1 shows the cross section of a *human* fingerprint replica, while Fig. 2b shows SEM images of the artificial fingerprints. They are similar but not exactly the same. Therefore, it is necessary to retain Supplementary Fig. 1. We have changed the figure legend in case of confusion.

Q3. Fig. 3e, which is the applied stimulus? Which is the measurement setup?

Response: We have added the missing information. The vibrational stimuli were applied to the sensors via a vibration generator at a pressure of ~ 10 kPa, and the corresponding vibration response signals were collected by an LCR meter.

Modification: On lines 1-3, Page 17, we have added “A vibration generator (Model BL-ZDQ-2185, Hangzhou Peilin Instrument Co. Ltd.) was used to apply constant frequency vibrations to the sensor at pressure of ~ 10 kPa, and the corresponding vibration response signals of the sensor were collected using the LCR meter.”

Q4. I suggest replacing Video 1 and Fig. 5 with experimental results at variable sliding velocity.

Response: Thanks for the reviewer’s suggestion. In the manuscript, we have replaced one of demonstrations of experimental results at a fixed sliding rate with a variable sliding rate (see the updated Fig. 5). In addition, we added a supplementary video showcasing the sensory system for texture recognition at variable sliding rates (see Supplementary Video 2).

Q5. The Discussion section looks as a conclusion section, the authors should revise the text accordingly to the section title.

Response: In the revised manuscript, we have removed the conclusive description in the “*Discussion*” section, and provided signal-to-noise ratio and achieved an effective number of bits of the sensor output signal.

Modification: On Lines 12-17, Page 14

“The fine fingerprint plays a key role to allow the sensor to fully interact with the fine features of textures, even at high sliding rates. Without the fingerprint, the recognition accuracy drops to only 54.5% at a sliding rate of $2 \text{ mm} \cdot \text{s}^{-1}$ (Supplementary Fig. 19). Furthermore, the slip-sensor exhibits a high signal-to-noise ratio of 86.79 dB and a high effective number of bits of 14.12 bits (Supplementary Fig. 20). These characteristics ensure that the sensor can precisely capture subtle tactile signals and deliver high-quality output in texture recognition.”

Q6. Page 13, lines 16-17. Which are the extracted signal features? The authors should introduce a Table with all the extracted and used features.

Response: In the revised manuscript, we have added a Table to show the top ten important extracted signal features (see the updated Supplementary Table 2).

Q7. The authors should evaluate and report in the paper, the signal-to-noise ratio and the effective number of bits of the sensor output signal.

Response: Thanks for the valuable suggestion. We have provided the signal-to-noise ratio and the effective number of bits of the slip-sensor in the “*Discussion*” section in the revised manuscript.

Modification: On Lines 14-17, Page 14

“.....Furthermore, the slip-sensor exhibits a high signal-to-noise ratio of 86.79 dB and a high effective number of bits of 14.12 bits (Supplementary Fig. 20). These characteristics ensure that the sensor can precisely capture subtle tactile signals and deliver high-quality output in texture recognition.”

REVIEWERS' COMMENTS

Reviewer #2 (Remarks to the Author):

The authors have addressed all my previous comments. I think the current version is with good quality to be accepted by Nature Communcations.

Response to reviewers for manuscript NCOMMS-23-15409C

Reviewers' comments

Reviewer #2 (Remarks to the Author):

The authors have addressed all my previous comments. I think the current version is with good quality to be accepted by *Nature Communications*.

Response: The authors appreciate the reviewer for the positive feedback on the revised version.